

# Airborne observations of newly formed boundary layer aerosol particles under cloudy conditions

Barbara Altstädter[1], Andreas Platis[2], Michael Jähn[3], Holger Baars[3], Janine Lückerath[4], Andreas Held[4], Astrid Lampert[1], Jens Bange[2], Markus Hermann[3], and Birgit Wehner[3]

[1]Institute of Flight Guidance, Technische Universität Braunschweig, Braunschweig, Germany
[2]Center for Applied Geosciences, Eberhard Karls University Tübingen, Tübingen, Germany
[3]Leibniz Institute for Tropospheric Research, Leipzig, Germany
[4]Atmospheric Chemistry, University of Bayreuth, Bayreuth, Germany

*Correspondence to:* Barbara Altstädter (b.altstaedter@tu-braunschweig.de)

**Abstract.** This study describes the formation of freshly formed boundary layer aerosol particles under classical called "non favourable" conditions by the existence of low-level strato cumulus clouds. Airborne measurements for vertical profiling atmospheric boundary layer properties and aerosol particles in the diameter range between 5 nm and 10 µm were repeatedly performed with the unmanned aerial system ALADINA (Application of Light-weight Aircraft for Detecting IN-situ Aerosol) at the research site of TROPOS (Leibniz Institute for Tropospheric Research) in Melpitz during three seasons between October 2013 and July 2015. More than 100 measurement flights were performed during this period on 23 different days with a total flight duration of 53 h. On 26 % of the measurement days, new particle formation was measured close to the inversion layer and was observed to be transported downwards during short time intervals of cloud gaps. However, the typical banana shape of new particle formation and growth was not seen at ground, only sporadic events appeared with low particle growth rate and thus might not have been classified as NPF by pure surface studies. This presentation focuses on two cases influenced by the passage of a low pressure system and minimal concentrations of $SO_2$ as indicator for precursor gases at ground; I) on April 4, 2014 by east wind and II) on June 21, 2015 under south-west wind. For a closure, an LES-model output was used for the second study in order to derive a full analysis of atmospheric boundary layer growth over the measurement site at small-scale with high spatial resolution. Pronounced turbulent fluxes of sensible and latent heat in the vertical distribution initiated fast mixing processes of freshly formed boundary layer aerosol. Furthermore, $CO_2$ and particle fluxes indicated downward transport and high deposition during sporadic nucleation events, so that NPF occurrence above with subsequent downward transport was clearly identified.

## 1 Introduction

The knowledge of atmospheric aerosols is still incomplete and thus contributes to the most significant uncertainties in climate model predictions, especially the aerosol sources, sinks and transformation processes (IPCC, 2007). Depending on their optical properties, aerosols interact significantly with solar radiation and hence influence the climate directly. Especially the formation of new particles due to nucleation by gas to particle conversion is subject of investigations. In this context, the formation and



growth rates of nucleation mode particles (ca. 1-15 nm) need a more profound understanding, as by subsequent and sufficient growth, the particles can act as cloud condensation nuclei and therefore influence the Earth's climate indirectly via affecting the hydrological cycle (e.g., Kerminen et al., 2005). Particle burst events were measured worldwide on different platforms like research stations for long-term monitoring, ships and airborne systems. Studies were performed at various altitudes from

the boundary layer up into the lower stratosphere and suggested different sources and proposed several mechanisms for new particle formation (e.g., Wiedensohler et al., 1996; Keil and Wendisch, 2001; Birmili et al., 2003; Kulmala et al., 2004; Hamed et al., 2010; Hamburger et al., 2011). According to these studies, new particle formation (NPF) is likely during preconditions of low temperature, high water vapour content, low surface of pre-existing particles that are often in relation to a low condensation sink, presence of precursor gases and high incoming solar radiation. However, further studies presented events of NPF under

non favourable conditions, likewise under the presence of clouds (e.g., Wiedensohler et al., 1997; Größ et al., 2015), and these investigations stress the hypothesis that nucleation is possible under a wider range of conditions than it has been expected so far.

In particular, the small-scale vertical distribution of aerosols in the atmospheric boundary layer (ABL) needs a more profound understanding and has to be implemented in models (Boy et al., 2006). A strong connection between the vertical variability

of aerosols and thermodynamic structures, caused by turbulence in the continental boundary layer, has been identified by Boy et al. (2003) at SMEARII (Station for Measuring Ecosystem-Atmosphere Relations) in Hyytiälä in Finland. The study suggested a connection between NPF and higher values of turbulent kinetic energy, in particular for so called "A-events" with high formation and growth rate of particles with a particle diameter of 3 nm. Nilsson et al. (2001) assumed that formation mechanisms are caused by dynamic processes in the mixed layer and entrainment zone. Further, Bigg (1997) suggested the

existence of breaking waves in the mixed layer as another possible mechanism for NPF, as breaking waves occurring in regions with high humidity and temperature fluctuations. Besides, Easter and Peters (1994) assumed favourables conditions near the inversion due to mixing processes that was afterwards demonstrated by Siebert et al. (2004) with the help of vertical profiling the continental ABL with a balloon-borne system. However, the measurements mentioned so far, except for the balloon, were conducted only in one fixed location so that the situation on a larger scale and in particular, a temporal development in different

altitudes is missing.

In this context, the use of airborne systems for atmospheric research is essential to deliver a detailed three-dimensional picture of the aerosol spatio-temporal distribution from the surface up to the free troposphere. The results of O'Dowd et al. (2009) showed the large-scale variability of the particle concentrations along air mass trajectories, and NPF and growth was observed over distances of several 100 km. For small-scale differences in atmospheric conditions (e.g. cloud occurrence), NPF

is distributed heterogeneously. The continental cloud top is favourable for NPF, furthermore the cloud-free regions in between of two cloud parcels (Keil and Wendisch, 2001).

The study of Wehner et al. (2015) presented the variety of nucleation mode particles around clouds over Barbados. The measurements were performed with the helicopter-borne measurement payload ACTOS (Airborne Cloud Turbulence Observation System) in the boundary layer. A high frequency of occurrence of nucleation events (around 83 % of all measurement days)

was observed, whereby 50 % of the events existed on a small-scale within 100 m in horizontal extent. In addition, the important




role of the entrainment zone for NPF was confirmed by the large data set of 91 flights and the influence of clouds for favourable NPF conditions was verified. Hamburger et al. (2012) studied atmospheric aerosols with the research aircraft FAAM BAe–146 and DLR Falcon 20 and suggested nucleation events in the ABL caused by the presence of high pressure systems and one event in the free troposphere around 8 km altitude due to the updraft during frontal passages. All in all, a high vertical variability

of atmospheric aerosols at different locations but in particular in the ABL was observed, however no detailed evolution of the vertical distribution of ABL aerosol in the same area.

This publication presents results of the unmanned aerial system (UAS) ALADINA that has already been extensively used for boundary layer aerosol field studies in Melpitz (Altstädter et al., 2015; Platis et al., 2016). Those previous investigations showed NPF in correlation with temperature and humidity fluctuations of several orders of magnitudes higher than in the remaining

part of the ABL. In addition, downwards transport of freshly formed particles to ground level was observed on a day of high incoming solar irradiance, therefore along a classical day for new particle formation. Furthermore, the downward transport of freshly formed particles was supported by the appearance of nucleation and particle growth and increasing concentrations of sulphur dioxide measured at ground.

Due to the high data set of vertical profiles that were performed during three different seasons with ALADINA at the research

site of TROPOS since 2013, a more complex role of new particle formation influenced by ABL processes is identified. In contrast to typical NPF events at ground by high incoming solar irradiance, other events were observed that might have not been considered by surface observations due to sporadic appearance of ultrafine particles. The publication benefits from the comparison of ALADINA with instrumentation at ground and one model output in order to get a better understanding of NPF conditions between surface and the free troposphere.

This article is structured as followed: Section 2 provides an overview of the unmanned research aircraft ALADINA and operation at the research site of TROPOS. An overview of the expanded version of the LES model output, that was used for a closure of ABL processes during the second case study, is presented in Subsect. 2.3. Section 3 shows the results of newly formed boundary layer aerosol under cloudy conditions during the two case studies, respectively. Finally, Sect. 4 concludes the presented study with the main goal to show the complexity and more frequent occurrence of NPF in the ABL, as expected so

far.

## 2   Instrumentation, model and research site for profiling boundary layer aerosol

### 2.1   Aerosol and meteorological payload on the unmanned research aircraft ALADINA

The aircraft Carolo P360 "ALADINA" (Application of Light-weight Aircraft for Detecting IN-situ Aerosol) was designed and developed for atmospheric research in order to investigate the vertical and the horizontal aerosol distribution depending on

atmospheric boundary layer properties. A detailed description of the airplane is given in Altstädter et al. (2015) and information on the new set-up can be found in Bärfuss et al. (2017). The payload on ALADINA is equipped with aerosol instrumentation and meteorological sensors with high temporal resolution.



The total aerosol particle number concentration is derived by two Condensation Particle Counters, CPCs (model 3007, TSI Inc., St Paul, USA), with different lower threshold diameters. In the first case study here, the cut-off sizes were 5 and 10 nm, respectively. The difference in the particle number concentrations of both CPCs ($N_5$ and $N_{10}$), in the following referred to as $N_{5-10}$, is used for the number concentration of freshly formed particles. During the second case study, the lower threshold

diameters of both CPCs were 7 and 12 nm ($N_7$, $N_{12}$; $N_{7-12}$), respectively. The CPCs were characterised to measure within an uncertainty of $\pm 20\,\%$ with a fast response time of 1.3 s.

In addition, an Optical Particle Counter, OPC (model GT-526, Met One Instruments Inc., Washington, USA), is installed and measures the size distribution of aerosol particles with six channels from 0.39 to 10 µm (ambient) in particle diameter with an uncertainty of $\pm 15\,\%$ and a temporal resolution of 1 s. Here, the aerosol particle size distributions are analysed in

the size windows between 390 to 700 nm, as larger particles were not relevant in the study due to minimal appearance. In the following, the particle size distributions of the three channels refer to the total aerosol particle number distribution in the size range between 390 and 500 nm ($N_{390}$), between 500 and 700 nm ($N_{500}$) and 500 to 700 nm ($N_{700}$).

The meteorological instruments are mounted at the tip of the aircraft nose next to the aerosol inlet. The sensor package consists of one five hole probe for measuring the three-dimensional wind vector with a temporal resolution of up to 40 Hz

and wind speed with an accuracy of $\pm 0.5\,\mathrm{m\,s^{-1}}$ (Wildmann et al., 2014a). The fast temperature sensors have a resolution of 10–20 Hz with an accuracy of $\pm 0.1$ K (Wildmann et al., 2013). Additionally, one humidity sensor is integrated that probes the water vapour content with a response time of 1.5 s with $\pm 3\,\%$ RH accuracy (Wildmann et al., 2014b).

## 2.2   Research site Melpitz and available instrumentation during experiments

The research site Melpitz of TROPOS (51° 32' N, 12° 56' E, 87 m a.s.l.) is located in the lowlands of Saxony, 41 km NE of

Leipzig, Germany, and surrounded by flat grass, agricultural areas and forests (e.g. Spindler et al., 2001; Spindler et al., 2004). The flat surface and the fact that no obstacle is in the direct vicinity of the station, enables the use as airfield for a safe take-off and landing of the UAS. Air masses arriving at Melpitz consist up to 60 % of originally maritime air, due to long distance transport by westerly winds and with predominantly enhanced concentrations of organic matter, sulphate and nitrate (Spindler et al., 2012). In the other frequent case, air masses are transported from industrial regions of the continental site of Eastern

Europe and therefore polluted with distinctly higher aerosol particle number concentrations in the boundary layer (e.g. Engler et al., 2007; Junkermann et al., 2016). Aerosol loads are in this case primarily characterised by anthropogenic emissions and classified as "near city background" by Putaud et al. (2004). Manninen et al. (2010) investigated NPF events at 12 different European sites including Melpitz within the framework of the EUCAARI (European Integrated project on Aerosol, Cloud, Climate, and Air Quality Interactions) project. During the intensive measurement period from March 2008 till April 2009,

57 % of the available analyses were classified as NPF event days with the major occurrence in warm seasons and a significant maximum in May.

All in all, the research site Melpitz of TROPOS offers a great potential for observing NPF within the ABL and inter-comparison of airborne data with monitoring at ground. A detailed description of meteorological sensors and gas analysers on site can be taken from Hamed et al. (2010). Sulphur dioxide concentrations were studied as main precursor gas at ground



in 1 min intervals. The temporal evolution of the aerosol at ground level was measured by a Twin Scanning Mobility Particle Sizer, TSMPS, with a 20 min scan between 3 nm and 800 nm in particle diameter (Wiedensohler et al., 2012). Hamed et al. (2010) focused on NPF events in connection with sulphur dioxide ($SO_2$) concentrations in a ten year period for the Melpitz site from 1996–2006. The fraction of NPF occurrence was 30 to 50 % and related to an increase of $SO_2$ concentrations. During the

earlier 1.5 y period between 1996 and 1997, 50 % of the available data was classified as nucleation with a maximum in June. However, only 30 % corresponded to NPF in the period from 2003 and 2006 with the highest frequency from June–September. To conclude, high occurrence of NPF in spring and summer were expected during airborne observations. In order to present a consistent analysis, the concentrations of precursor gases are studied that were measured at ground station with the major role of anthropogenic $SO_2$ for new particle formation.

In order to classify the diurnal cycle of clouds in the vertical structure and top of the ABL, ceilometer und lidar data were used. In addition, EC data of particle fluxes was used for the second case study in order to observe the vertical transport processes.

## 2.3 Large-eddy simulations with forced mescoscale model output for MelCol 2015

ALADINA was operated during the field study Melpitz Column (MelCol) from June 16 until July 1, 2015. For the second case

study on June 21, 2015, an LES model output was available and is used for ABL description in order to derive a continuous data set of the vertical distribution of latent heat fluxes ($w'q'$), sensible heat fluxes ($w'\theta'$) and turbulent kinetic energy (TKE). The model itself is described in the following.

Simulations associated with the MelCol 2015 measurement campaign are performed with the All Scale Atmospheric Model (ASAM, Jähn et al., 2015). It has recently been used to investigate heat island effects on atmospheric boundary layer modifi-

cation, cloud initiation and vertical tracer mixing in the trade wind regime (Jähn et al., 2016). For the present study, large-eddy simulations (LES) for selected days during the campaign were performed with respect to the present synoptic situation, including changes due to large-scale advective tendencies and incoming radiation.

The computational domain is $25.2 \times 25.2 \, km^2$ wide with Melpitz field site located in the domain centre. Since the focus lies on atmospheric boundary layer, the model top is set to a height of 4 km. The land use around Melpitz is characterised by

mainly different forest types (farm land, shrubland, urban areas), see Fig. 1. Due to the relatively flat environment, orographic structures are not taken into account for the simulations, i.e., the model domain could be chosen as flat surface.

The model physics is described by the prognostic TKE equation (Deardorff, 1972; Moeng and Wyngaard, 1989), by the two-moment microphysics scheme (Seifert and Beheng, 2006) with excluded ice phase in order to save computation time and by the multi-layer soil and land-use model (TERRA_ML, Doms et al., 2011) with a revised scheme for the surface layer (Jimínez

et al., 2012).

The utilised model set-up is similar to the one described in Heinze et al. (2017), and it includes large-scale forcing tendencies due to advection of heat and water vapour, subsidence from COSMO reanalysis (Heinze et al., 2017); direct, diffuse and terrestrial radiation (1 min averages) and soil data (temperature and moisture) from the German weather service (Deutscher Wetterdienst, DWD) station "Klitzschen bei Torgau", which is located 3 km away from the Melpitz field site.





## 3 Results and discussion

### 3.1 Case I April 2014

In this section, NPF observations are shown on April 4, 2014, as this day is in contrast to previous investigations from Platis et al. (2016) under clear sky conditions. Large scale analysis revealed that air masses were transported initially from Saharan regions over Eastern Europe within the last days via north-easterly winds. According to data of surface pressure systems that are publicly available by DWD, the research station was influenced by a low pressure system and especially by the passage of a cold front that finally occluded around 18:00 UTC (-02:00 MEZ).

A typical "banana shape" (Heintzenberg et al., 2007), as a consequence of adequate condensation on precursor gases, was not observed this day. The development of the size distribution at ground level was calculated from TSMPS (see Subsect. 2.2) data from the research site. The total aerosol particle number concentration in the diameter range of 10 to 800 nm was rather low with a mean concentration of $8 \times 10^3 \, \text{cm}^{-3}$ in Melpitz on April 4, compared to other days in spring time (e.g., Hamed et al., 2010; Platis et al., 2016). Particles belonging to a diameter range between 100 and 200 nm were predominant with a mean total aerosol particle number concentration of $1.0 \times 10^4 \, \text{cm}^{-3}$ in $\text{dN dlogD}_p^{-1}$ and evenly distributed during the whole day. Larger particles with a particle diameter from 200 to 800 nm played a minor role with a few hundred particles per $\text{cm}^3$. Only the ultrafine particles with 10 to 100 nm particle diameter could be distinguished in the temporal distribution and were clustered in two events in the early morning at 02:00 and 04:15 UTC, also one further event in the afternoon at 14:00 UTC (Fig. 2a). A fourth sporadic occurrence of particles in the diameter range between 80 and 150 nm was detected at around 19:00 UTC and reached a total maximum aerosol particle number concentration of $1.5 \times 10^4 \, \text{cm}^{-3}$.

In addition, gas concentrations of $SO_2$ and $NO_x$ are presented in Fig. 2b. The $SO_2$ concentration varied between 1.4 and 4.4 $\mu\text{g m}^{-3}$ from the early morning until 10:00 UTC. Afterwards, the concentration increased slowly to a total maximum of 11 $\mu\text{g m}^{-3}$ at 20:40 UTC and decreased to 4.8 $\mu\text{g m}^{-3}$ in the night. The values of $NO_x$ varied between 10.2 and 34.6 $\mu\text{g m}^{-3}$ during the day, with temporary higher concentrations in the early morning at 03:50 UTC, from 07:55 to 09:40 UTC and in the afternoon around 18:00 UTC.

The temperature was low between 5.3 and 12.3 °C and the relative humidity reached high values up to 98.1 % in the early morning with a maximum at 08:25 UTC. Due to specific more and only short periods of clear sky, the global radiation, $G$, varied between 400 and 720 $\text{W m}^{-2}$ (Fig. 2c). The wind speed was moderate in the range of 2 and 4 $\text{m s}^{-1}$. Besides one sharp change of the wind direction, $dd$, from north to south-west between 08:15 and 08:55 UTC, the prevailing wind direction was from north-east (Fig. 2c).

### 3.1.1 Cloudy conditions and heterogeneously mixed atmosphere

A heterogeneous lower atmospheric structure was identified with the ceilometer data and the results for the lowermost 4 km are presented in Fig. 3. A stable night time ABL with a depth of less than 600 m was observed from midnight until around 07:00 UTC. Dense ABL clouds formed within this humid layer from about 07:00 to 10:00 UTC. After 10:00 UTC, the clouds dissolved and the humid convective boundary layer started growing. The maximum ABL depth was observed at around 14:30



UTC with an ABL-top of 800 m. Convection decayed and a residual layer (RL) remained after 16:00 UTC. Various particle (shown as green and yellow colours) and cloud (shown in white) layers were observed at different altitudes during the whole day. Partly, the lofted aerosol layers (steady existent up to 1.5 km) were mixed into the ABL during the growth process. The higher aerosol layers above 1.5 km height were also present during the whole day but did obviously not affect the boundary

layer aerosol conditions. After 20:00 UTC, precipitation was observed which did not reach the ground.

### 3.1.2   NPF in the vertical distribution

In the following, vertical profiles obtained with the UAS ALADINA are shown and the flight times are summarised and connected with weather conditions and gas concentration of $SO_2$ in Tab. 1. During Case I, on April 4, six measurement flights were performed between a maximum height of 700 and 1000 m from 06:15 until 13:58 UTC. Figure 4 displays four vertical

profiles, from left to right, of potential temperature $\theta$, water vapour mixing ratio $q$, total aerosol particle number concentration measured with two CPCs in the particle diameter of 5 nm (red line) and 10 nm (blue line), respectively, and the total aerosol particle number concentration observed with the OPC in the particle diameter of 390 nm (pink line), 500 nm (green line) and 700 nm (turquoise line) in the time interval at (a) 09:06 UTC, (b) 10:45 UTC, (c) 11:47 and (d) 13:50 UTC.

    The first profile was taken at 09:06 UTC (Fig. 4a) and showed a strong inversion layer in the height between 420 and

550 m a.g.l. that was influenced by air masses of high moisture, identified by the rapid increase of $q$ from 2 to 18.5 g kg$^{-1}$ within the inversion layer. $N_5$ decreased continuously from $6.0\times10^3$ to $4.0\times10^3$ cm$^{-3}$ between the height of 100 and 420 m a.g.l. Above, NPF event was observed, shown by $N_{5-10}=3.5\times10^3$ cm$^{-3}$ that was strongly connected to the layer of high moisture. Above the inversion, the total aerosol particle number concentration decreased to $3.0\times10^3$ cm$^{-3}$ for the particles measured with both CPCs and remained constant to the height of 700 m. In case of the OPC data, larger particles were mixed

below inversion layer and significantly decreased above. At ground, less than 250 cm$^{-3}$ were measured with the 390 nm channel, 50 cm$^{-3}$ for the particle diameter of 500 nm and larger particles (700 nm to 5 µm) were not detectable. Up to the inversion layer at 420 m a.g.l., the vertical distributions of $N_{390}$ and $N_{500}$ were constant and decreased rapidly along with the maximum of $N_{5-10}$. Between the altitude of 470 and 700 m a.g.l., the total aerosol particle number concentration of $N_{390}$ and $N_{500}$ were constant with low values of $N_{390}=110$ cm$^{-3}$ and $N_{500}=20$ cm$^{-3}$.

At 10:45 UTC (Fig. 4b), the inversion layer was still detectable and lifted up to the height between 500 and 700 m a.g.l. and connected to the layer of high moisture. Here, only data from the CPC with the lower cut-off size of 5 nm were available. The CPC detected particles up to $0.8\times10^4$ cm$^{-3}$ between ground level and the capping inversion at the height of 450 m a.g.l. Hereafter, the amount of $N_5$ increased steadily to $3.8\times10^4$ cm$^{-3}$ at the altitude around 700 m a.g.l. with two distinct layers of particles. Below the inversion, ultrafine particles were evenly dispersed within the BL, but above the inversion a heterogeneous

distribution was observed up to the height of 800 m a.g.l., where the total aerosol particle number concentration decreased rapidly to a minimum of $0.5\times10^4$ cm$^{-3}$. At the same time, particles belonging to the diameter of 390 nm were also evenly distributed below the inversion and reached values around 220 cm$^{-3}$. Within the height of 450 and 500 m, the concentration decreased to $N_{390}=130$ cm$^{-3}$. Between 500 and 950 m in altitude, the total aerosol particle number concentration varied





between 110 and 120 cm$^{-3}$. The same distribution occurred for particles in the diameter size of 500 nm, but with minimal concentrations.

The next vertical profile was taken at 11:47 UTC (Fig. 4c). A high increase of $N_5$ with a maximum total aerosol particle number concentration of $3.8 \times 10^4$ cm$^{-3}$ was observed in the height between 150 and 250 m a.g.l. A second layer with enhanced

values of $N_5$ was detected above the inversion layer in the heights of 630 and 950 m a.g.l. with a maximum of $3.4 \times 10^4$ cm$^{-3}$. The results of the OPC were almost the same as 1 h before and significantly affected by the inversion layer.

Later, at 13:50 UTC (Fig. 4d), the total aerosol particle number concentration was detected with both CPCs again. From ground level up to the inversion at 800 m a.g.l., the maximum of N$_{5-10}$=$0.5 \times 10^4$ cm$^{-3}$ appeared. Above 800 m a.g.l., the decline of N$_{390}$ and N$_{500}$ was present and linked to an increase of ultrafine particles.

During the temporal development of ABL aerosol, the lifted layer of freshly formed particles was transported downwards, which can be further seen by the temporal appearance of the small particles of a few nm in diameter in the aerosol data at ground level (Fig. 2a). Nevertheless, the particle growth was not sufficient or interrupted by other processes, so that a typical formation event like a "banana shape" could not be identified by ground data. Regarding OPC observations, accumulation mode particles were still homogeneously mixed in the ABL up to the inversion. To sum up, vertical profiles showed the strong

dependence of airborne measured particles on the structure of the ABL. The overall observation was the occurrence of NPF connected to a layer of high moisture close to the inversion depending on the decline of N$_{390}$ and N$_{500}$, as an indicator for the condensation sink in that altitude that might have favoured the particle formation process. The same behaviour was observed by Rose et al. (2015), who suggested NPF appearance by highest condensation sink in the transition zone between ABL and free troposphere (FT).

## 3.2  Case II June 2015

The second case presents observations from June 21, 2015. Within the last five days starting on June 16, 2015, retrieved back-trajectories showed air masses originated over the Atlantic, so that a low aerosol load was expected. Further, the research site was influenced by a low pressure system that led to a mixed structure of strato cumulus (StCu) clouds in the height of 500–2500 m. The cloud coverage can be further taken from the Polly$^{XT}$ lidar (e.g., Althausen et al., 2009; Baars et al., 2017) data

from 06:00 until 18:00 UTC (Fig. 5).

An overview of the performed measurement flights with ALADINA in connection with weather conditions and gas concentrations of SO$_2$ between 0.8 and 3.7 µg m$^{-3}$ can be found in Tab. 2. In this case, seven flights were performed between 08:00 and 15:32 UTC with a total maximum height of 1200 m. The temperature at 1 m height was in the range of 9.4 and 18.3°C during the day. The wind speed was weak between 0–3.7 m s$^{-1}$ coming from SW, so that nucleation was not expected by clean

air masses, according to the results of Junkermann et al. (2016).

The size distribution measured by TSMPS at ground are displayed in Fig. 6a. The aerosol load was constantly dispersed until 07:20 UTC with the highest total aerosol particle number concentrations of $5.0 \times 10^4$ cm$^{-3}$ in the particle diameter of around 20–50 nm. Particles belonging to the accumulation mode were insignificant on this day and reached values of only a few 100 particles cm$^{-3}$. At 07:50 UTC, a sporadic NPF event was observed with a maximum total aerosol particle number





concentration of $1.5 \times 10^5 \, \mathrm{cm}^{-3}$ in the particle diameter range of 3 and 10 nm. At the same time, a significant decline of accumulation mode particles was measured. However, the new particle formation event dissolved within 20 min and the total aerosol particle number concentration of particles larger than 20 nm increased, although the concentrations were still low during the day. In the following, five events of a significant increase of the total aerosol particle number concentration in the

diameter range between 7 and 20 nm were temporarily clustered at 10:10, 11:50, 13:20, 14:50 and 16:10 UTC. The sporadic formation events were ongoing with a decrease of total aerosol particle number concentration of particles in the diameter of 20–50 nm. Figure 6b displays the diurnal cycle of $CO_2$ fluxes, $F_{CO_2}$. During non events, the fluxes were positive and reached a maximum of $0.2 \, \mathrm{mg \, kg^{-1} \, m \, s^{-1}}$ in the night at 01:30 UTC. At 05:00 UTC, negative values of $F_{CO_2}$ occurred with a maximum of $-0.2 \, \mathrm{mg \, kg^{-1} \, m \, s^{-1}}$ at 08:40 UTC, that was at the same time with the sporadic formation event. In order to show a direct

dependence of water vapour on nucleation, Fig. 6d depicts the liquid water path, LWP, during the measurement period. Highest values of up to $10 \, \mathrm{kg \, m^{-2}}$ indicated the existence of dense ABL clouds and nucleation occurred in short time period by mean LWP of $3.5 \, \mathrm{kg \, m^{-2}}$.

Further, Fig. 7 displays particle fluxes taken from EC (eddy-covariance) on the same date. During sporadic formation events, significant deposition occurred, taken from the negative values of particle fluxes. In contrast, in between formation events,

emission was observed, indicated by positive particle fluxes.

Again, no typical shape of new particle formation with high and steadily increasing growth rate was observed during this day with dense cloud coverage of strato cumulus, however nucleation appeared sporadically by downwards transport that was indicated by negative fluxes of $CO_2$ and deposition of particle fluxes.

### 3.2.1 ABL properties during the NPF event

Atmospheric boundary layer conditions were not available from the UAS and estimated with the LES-model output (Sect. 2.3) in a vertical resolution of 50 m, beginning at the altitude of 25 m. The top of the ABL height varied between 900 to 1200 m in the period of 06:00 until 16:00 UTC (Fig. 8). The water vapour mixing ratio $q$ increased during the day in the vertical distribution between 0 and 1500 m. The maximum of $18.5 \, \mathrm{g \, kg^{-1}}$ was estimated in the afternoon in the 1 h time interval between 14:00 and 15:00 UTC in the height of 1200 m. The vertical profile of turbulent kinetic energy, TKE, from the surface up to 1500 m showed

a strong connection with the structure of the ABL. In the lowermost 50 m, TKE reached highest value and a total maximum of $1.18 \, \mathrm{m^2 \, s^{-2}}$ between 06:00 and 07:00 UTC. At 150 m, TKE decreased compared to the surface inversion and was stable between 1200 and 1500 m. Close to the transition zone into the FT, TKE was negligible. All in all, the LES output showed a capped inversion with high moisture and moderate wind speed in the lowermost 200 m and simultaneously the highest value of TKE.

### 3.2.2 Vertical mixing of NPF

Two of the six sporadic nucleation events (see Fig. 6) were observed with the UAS ALADINA on June 21, 2015. The first case corresponds to the NPF event at around 08:00 UTC and the results are shown in Fig. 9. The vertical profile of sensible heat flux $w'\theta'$ and latent heat flux $w'q'$ were estimated with the LES model and presented for an 1 h interval between 08:00 and





09:00 UTC from the surface up to 1000 m in altitude. During this period, three vertical profiles of the UAS were performed at 08:13 UTC (solid line), 08:20 UTC (dotted line) and 08:26 UTC (dashed line) in the height between 100 and 950 m. Note, during this study, the lower cut-off sizes of the two CPCs were 7 and 12 nm in the particle diameter, respectively (see Subsect. 2.1). At 08:13 UTC (black line), the vertical profile of $N_{7-12}$ showed a high variance; at 100 m the total aerosol

particle number concentration was $8 \times 10^4$ cm$^{-3}$ and decreased to $3 \times 10^4$ cm$^{-3}$ at the height of 390 m. Above, a significant maximum of $16 \times 10^4$ cm$^{-3}$ was measured at the height of 500 m and existed up to 790 nm. Between the height of 800 and 950 m, $N_{7-12}$ decreased to the total aerosol number concentration of $10 \times 10^4$ cm$^{-3}$ in connection to a layer of maximum $w'q'$=25 g kg$^{-1}$ m s$^{-1}$ and negative values of $w'\theta'$ from -2 to -18$\times 10^{-3}$ K m s$^{-1}$. Only 17 min later, at 08:20 UTC, the lifted layer of enhanced aerosol concentration above 500 m was not observed anymore. However, below the altitude of 410 m, the

vertical distribution of $N_{7-12}$ was as before. At 08:26 UTC, the total aerosol particle number concentration of $N_{7-12}$ was homogeneously mixed in the vertical distribution with a mean total aerosol number concentration of $2.5 \times 10^4$ cm$^{-3}$.

In order to validate the results of airborne data, integrated total aerosol number concentrations in the same particle diameter between 7 and 12 nm of the TSMPS are given. At 07:40 UTC, $N_{7-12}$=$2 \times 10^4$ cm$^{-3}$ was measured at surface, before nucleation started. During the first flight of ALADINA, the maximum of $N_{7-12}$ was $9 \times 10^4$ cm$^{-3}$ at ground. Further, $N_{7-12}$ decreased to

$3 \times 10^4$ cm$^{-3}$ at 08:30 UTC) and $1 \times 10^4$ cm$^{-3}$ at 08:50 UTC. ALADINA observations in the lowermost 100 m were consistent with TSMPS data at ground. Thus, ground observations calculated by integration a size scan over 20 min can not reproduce the significant spread of $N_{7-12}$ within the ABL, but are still consistent with profiling data.

Figure 10 displays the same parameters as given in Fig. 9 for the time interval between 14:00 and 15:00 UTC, corresponding to the fifth sporadic nucleation event seen in TSMPS observations on June 21, 2015. At 14:16 UTC, the vertical

profile of ALADINA (solid line) showed a homogeneous distribution of $N_{7-12}$ with a mean total aerosol particle number concentration of $3.5 \times 10^4$ cm$^{-3}$. The second profile was measured at 14:57 UTC (dotted line) with a significant increase of $N_{7-12}$=$1.8 \times 10^5$ cm$^{-3}$ at ground and decreased rapidly in the vertical pattern between the height of 160 and 1100 m. The region corresponds to a maximum of sensible heat flux of $56 \times 10^{-3}$ K m s$^{-1}$. The third vertical profile was taken at 15:06 UTC (dashed line) and $N_{7-12}$ was mixed in the vertical distribution by a mean total aerosol number concentration of $4.2 \times 10^4$ cm$^{-3}$.

The vertical distribution of latent heat flux showed an continuously increase with a maximum of 38 g kg$^{-1}$ m s$^{-1}$ at 1150 m.

Again, airborne data was consistent with ground observations by TSMPS, however the temporal evolution of vertical profiles led to the assumption, that NPF occurred at ground instead of aloft. However, fast mixing was observed and at the same time an increase of the total aerosol particle number concentration in the vertical pattern.

## 4  Concluding remarks

Freshly formed boundary layer aerosol was measured with the unmanned aerial system ALADINA under cloudy conditions at the research site of TROPOS in Melpitz. In total, 105 airborne measurements with total flight duration of 53 h were performed on 23 measurement days since October 2013. During six of these measurement days, new particle formation (NPF) events were observed under non favourable conditions (e.g., low concentrations of precursor gases, dense cloud coverage) near the





inversion layer and were mixed vertically induced by atmospheric boundary layer (ABL) dynamic processes. However, only sporadic nucleation events were detected at ground and might not have been taken into account as NPF event.

   This study focused on two different cases: On April 4, 2014, one formation event was captured above the inversion layer at 420 m a.g.l. in relation to a significant moist layer with a water vapour mixing ratio of $18.5\,\mathrm{g\,kg^{-1}}$ after a cold front passage. Further, the NPF event was linked to a condensation sink of larger particles belonging to the accumulation mode at the same altitude. The results are consistent with the study of Rose et al. (2015), where NPF events were mainly observed during periods with a high condensation sink, not during those with concentrations of high sulphuric acid between the atmospheric boundary layer (ABL) and free troposphere. Ceilometer backscatter signal showed that the atmosphere was heterogeneously mixed with various layers of enhanced concentrations of atmospheric aerosols and several cloud levels during the whole day. Especially from 08:30 to 10:30 UTC, a dense cloud coverage at the altitude around 400 m existed. The layer of the freshly formed aerosol spread out vertically in the boundary layer with $3.5\times10^3$ particles $\mathrm{cm^{-3}}$ between 5 and 10 nm in particle diameter and reached a maximum total number concentration of $3.8\times10^4\,\mathrm{cm^{-3}}$ for particles with a diameter size exceeding 5 nm. Ground aerosol was simultaneously investigated with a TSMPS, however ground observations did not catch the newly formed boundary layer aerosol. The research site was influenced by NE wind, so that high emissions of anthropogenic sources were expected from Eastern Europe like stated by Junkermann et al. (2016), who found new particle formation events in connection with local plumes over Eastern Germany. However, changes in wind direction were not seen in wind measurements with ALADINA. Nevertheless, the cold front passage might have led to occurrence of different air masses with anthropogenic emissions transported into the ABL. The study of Bianchi et al. (2016) at the Jungfraujoch site in the free troposphere support the present observations; NPF was not related to sulfuric acid formation depended on high concentrations of organic compounds, e.g. CO so that anthropogenic sources may be relevant.

   On June 21, 2015, the site was influenced by a low pressure system with strato cumulus clouds and SW wind. At 08:13 UTC, ALADINA measured a NPF event above 450 m a.g.l. in a vertical distribution of high humidity. The maximum of total aerosol particle number concentration was $1.6\times10^5\,\mathrm{cm^{-3}}$ at the altitude of 420 m a.g.l. and increased rapidly while descending. Within 17 min, the newly formed aerosol load was mixed vertically or disappeared as confirmed by pronounced fluxes of sensible heat with a maximum of $81\times10^{-3}\,\mathrm{K\,m\,s^{-1}}$ in the lowermost 160 m. The temporal evolution showed that the formation observed at ground originated from the ABL and was mixed vertically and transported downwards to the surface, as previously observed in Platis et al. (2016) under clear sky conditions. Another sporadic nucleation event was captured with the UAS in the afternoon between 14:10 and 15:10 UTC and showed the formation process occurred at ground and was rapidly mixed in the vertical distribution initiated by significant increase of sensible heat flux with a maximum of $58\times10^{-3}\,\mathrm{K\,m\,s^{-1}}$ in the lowermost 150 m.

   Besides, the UAS observations were consistent with TSMPS data in the same diameter range of $N_{7-12}$, clarifying the reliability of the system. In addition, surface observations of $CO_2$ fluxes and particle fluxes showed the high variability during nucleation events by downdraft and deposition, in contrast to updraft and emission during short periods of non-events by dense cloud coverage.

   To conclude, high water content, low surface of pre-population and pronounced fluxes of latent heat, sensible heat and turbulent kinetic energy were clearly the dominant factor in the atmospheric boundary layer to make occur NPF, even if





conditions were generally not favourable in contrast to previous study of Platis et al. (2016) during clear sky conditions. To summarise, the observations lead to the assumption, that occurrence of NPF within the boundary layer are underestimated, if only ground observations are available and a more profound understanding of the vertical structure of the ABL is necessary in order to characterise NPF events.

5  *Code and data availability.*  At the current state, the data sets are not publicly accessible, as analyses and further publications of the campaign MelCol 2015 are still in progress by other participants, but will be delivered upon request. The open access of data in e.g. PANGAEA will be provided in future. To get access to the source code and additional scripts for pre- and postprocessing of LES model, registration at the TROPOS Git hosting website gitorious.tropos.de/ is mandatory. Additional information can be found at the ASAM webpage (http://asam.tropos.de).

10  *Competing interests.*  The authors declare that they have no conflict of interest.

*Acknowledgements.*  This work is funded by the German Research Foundation (DFG) under the project numbers LA 2907/5-1, WI 1449/22-1, BA 1988/14-1 and LA 2907/5-2, WI 1449/22-2, BA 1988/14-2. The authors thank Lutz Bretschneider, Nico Weil and Simon Nieschke for their technical support and for performance as good safety pilots. We acknowledge Norman Wildmann, Henning Busse and Andreas Scholtz for active assistance with the ALADINA system. Thanks to Huguette Djoumsap Takam, Mareike Decker and Maike Siekmann who

15  were studying different formation events in Melpitz and have brought forward this study. A special thank to the whole TROPOS-team and MelCol participants, especially Ralf Käthner, Achim Grüner and Gerald Spindler for the valuable support on site and for offering the access to Melpitz and TROPOS data. The authors thank Armin Raabe of the Institute of Meteorology from the University Leipzig for providing turbulence data at ground. The simulations were performed at the Center for Information Services and High Performance Computing (ZIH) at TU Dresden.




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



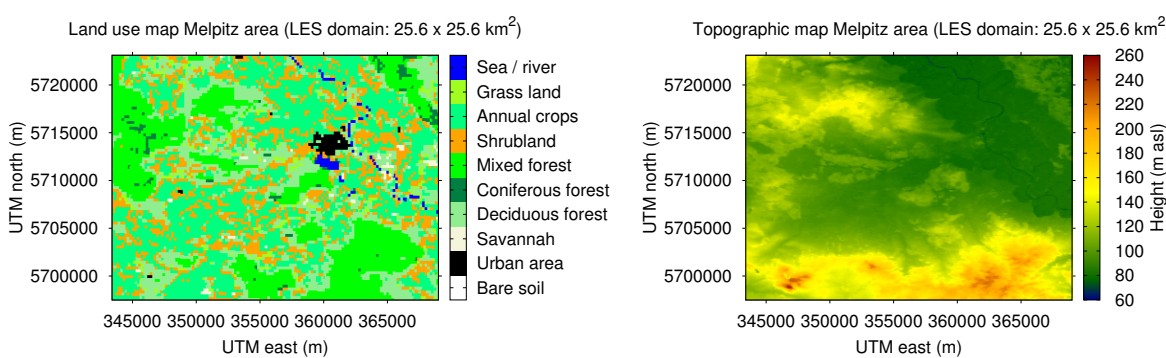

**Figure 1.** Land use and topography of the simulation domain around the Melpitz field site. The large variation of vegetations on small-scale are indicated by different colours (left) and the topography ranging from 60 to 260 m above sea level (a.s.l.) is also indicated with the same resoultion (right).





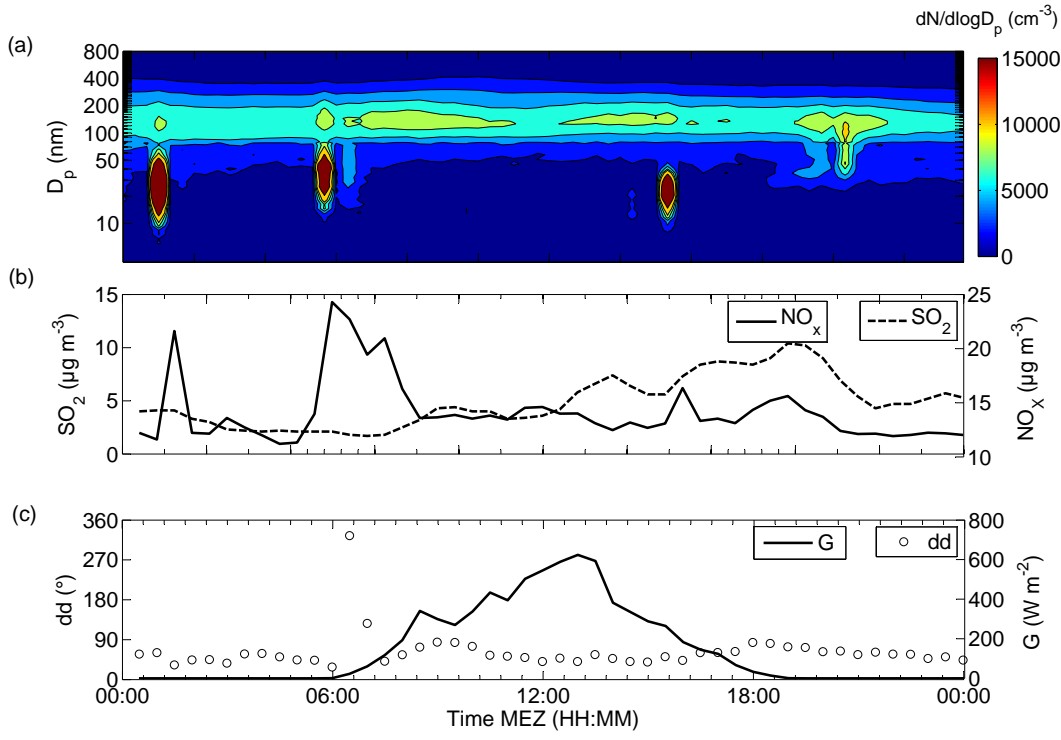

**Figure 2.** Ground observations in Melpitz on April 4, 2014. (a) Particle size distribution measured with TSMPS in the particle diameter range $D_p$ from 3 to 800 nm, (b) gas concentrations of $SO_2$ and $NO_x$, (c) wind direction $dd$ and global radiation $G$.





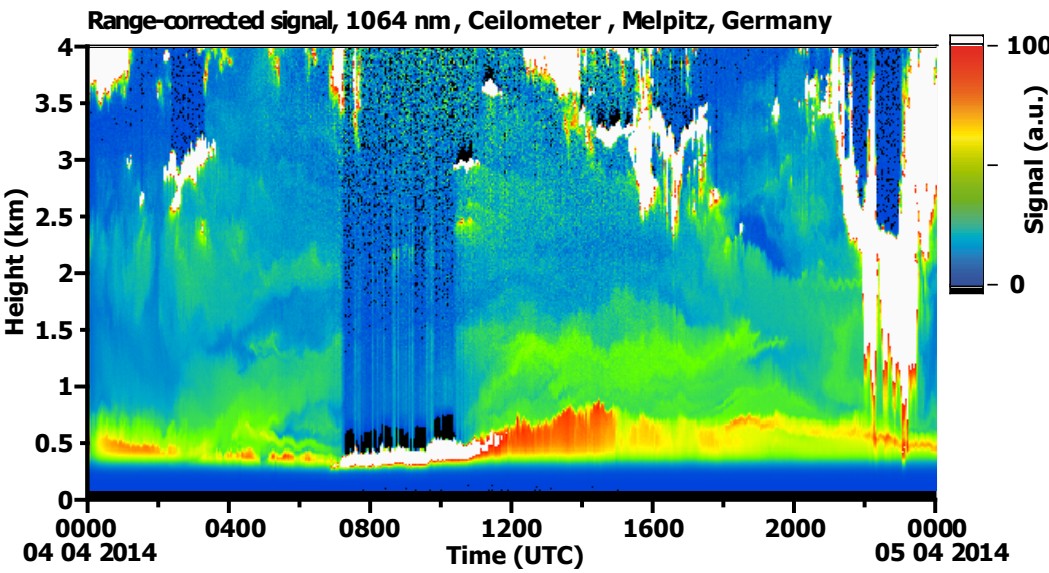

**Figure 3.** Backscatter signals (range corrected at the wavelength of 1064 nm) of the ceilometer installed at Melpitz on April 4, 2014. White parts show clouds and colour scales in green up to red the existence of atmospheric particles.





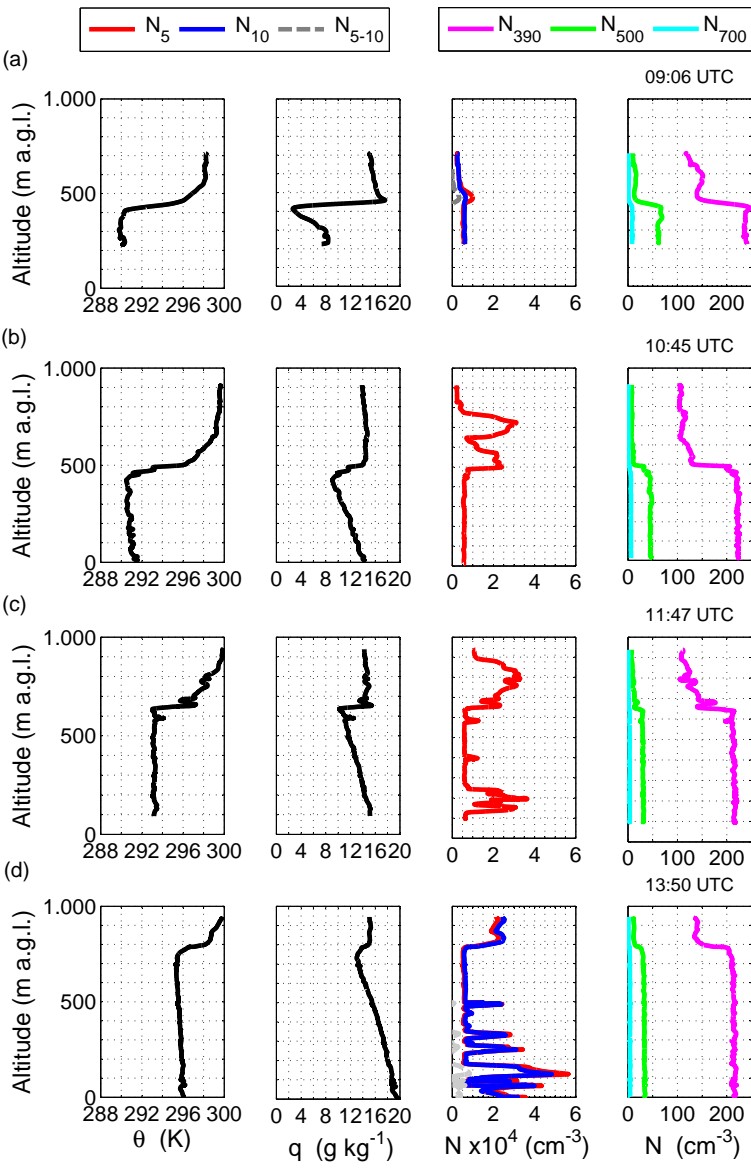

**Figure 4.** Vertical profiles of potential temperature $\theta$, water vapour mixing ratio $q$, total aerosol particle number concentration of ultrafine particles measured with two CPCs in the particle diameter above 5 nm (red) and above 10 nm (blue) and total aerosol particle number concentration measured with OPC in the particle diameter of 390 nm (pink), 500 nm (green) and 700 nm (turquoise). All data were measured with ALADINA at (a) 09:06, (b) 10:45, (c) 11:47 and (d) 13:50 UTC on April 4, 2014.





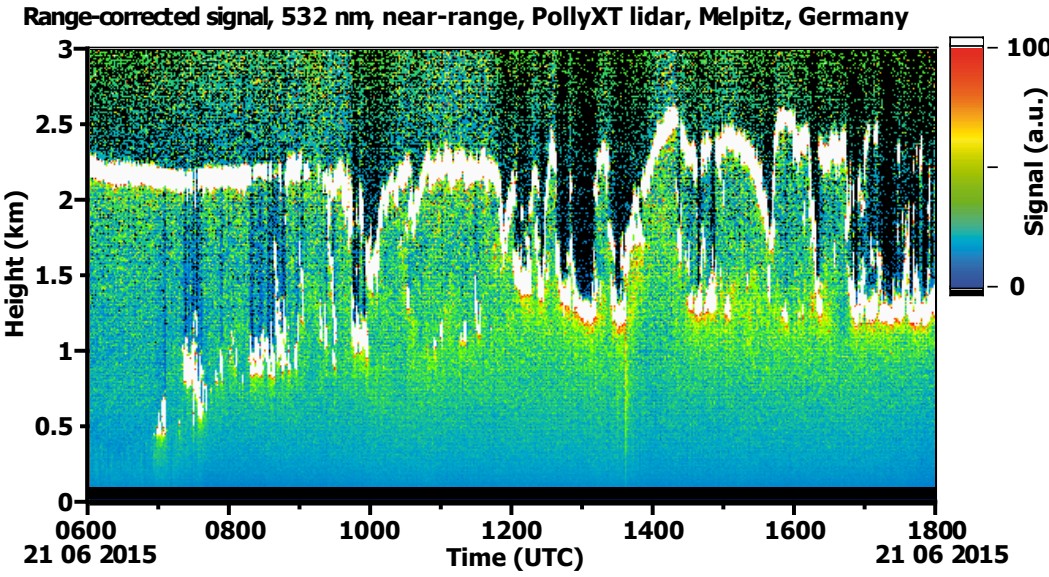

**Figure 5.** Backscatter signals (range corrected at the wavelength of 532 nm) of Polly$^{XT}$ lidar installed at Melpitz on June 21, 2015. White parts show clouds and colour scales in green up to red the existence of atmospheric particles.





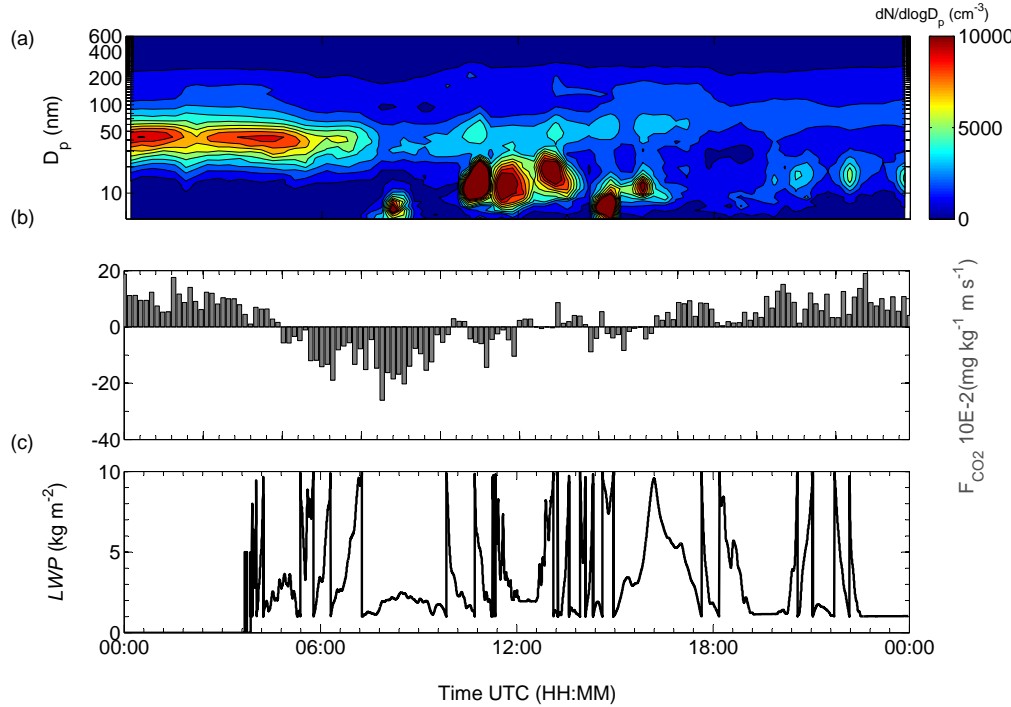

**Figure 6.** (a) Particle size distribution measured with TSMPS, (b) fluxes of $CO_2$, $F_{CO_2}$, and (c) liquid water path, LWP, during MelCol experiment on June 21, 2015. The homogeneous aerosol load was affected by ABL growth after 07:30 UTC and six sporadic formation events (seen in red dots in the size range of 10 to 20 nm) were indicated during daytime between 08:10 and 16:10 UTC under cloudy conditions and SW wind.





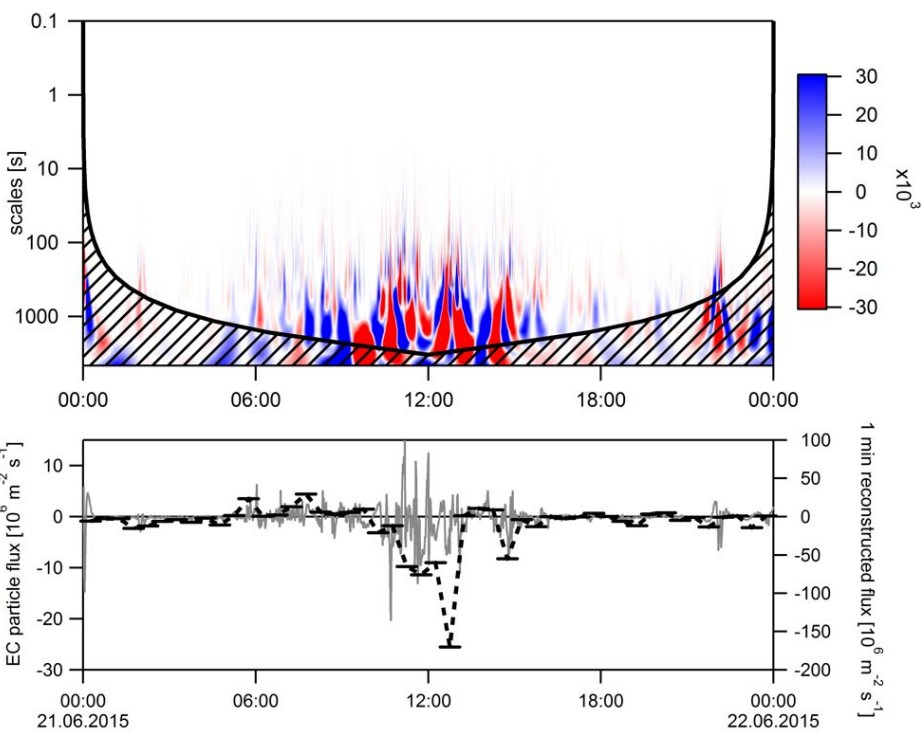

**Figure 7.** Wavelet analyses of EC station in time scale on top and below particle fluxes estimated during MelCol experiment on June 21, 2015. During sporadic new particle formation, deposition was observed (red), in contrast to emission by non-event sections (blue). The 30 s average of EC particle fluxes (dashed line) showed the significant deposition affected by NPF at 12:30 UTC.





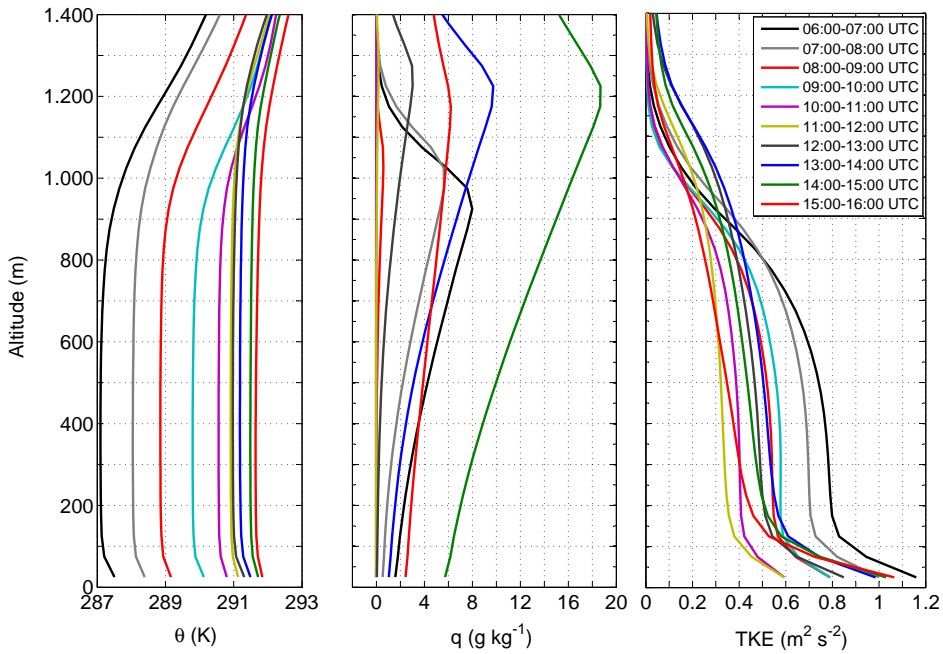

**Figure 8.** Potential temperature $\theta$, water vapour mixing ratio $q$ and turbulent kinetic energy TKE depending on the altitude, estimated with the LES model in 1 h time intervals starting from 06:00 until 16:00 UTC during the MelCol experiment on June 21, 2015. Highest values of $q$ were estimated between the height of 950 and 1210 m close to inversion layer, respectively and appeared before sporadic NPF occurrence at ground.





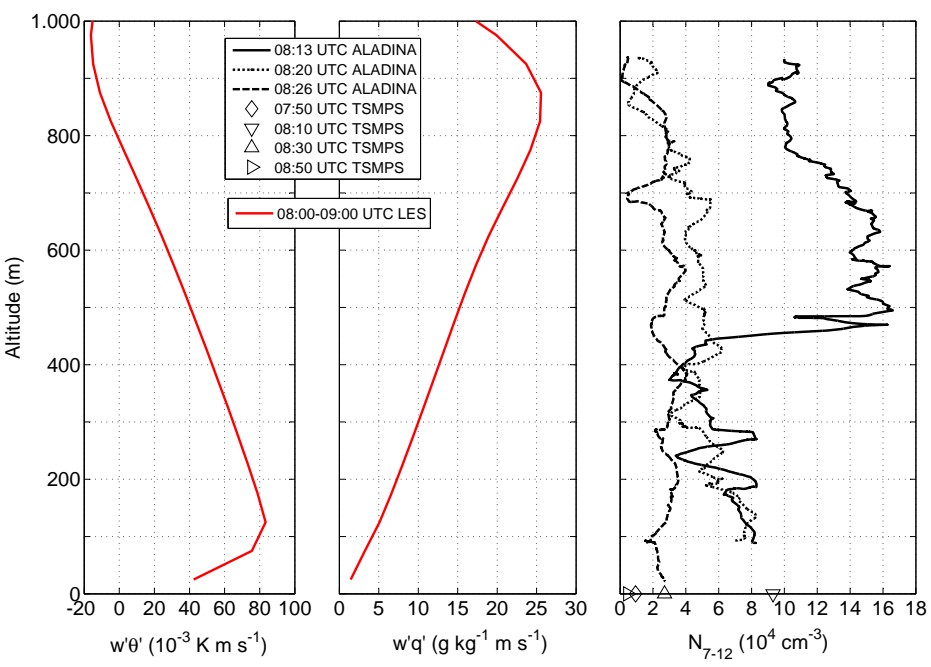

**Figure 9.** Vertical profiles of sensible heat flux $w'\theta'$ in $\mathrm{K\,m\,s^{-1}}$ and latent heat flux $w'q'$ in $\mathrm{g\,kg^{-1}\,m\,s^{-1}}$ obtained by LES output in the interval of 08:00–09:00 UTC (red lines). Besides, three vertical profiles of freshly formed boundary layer aerosol in the particle diameter between 7 and 12 nm ($N_{7-12}$) measured with the CPCs on the UAS ALADINA at 08:13 UTC (solid line), 08:20 UTC (dotted line) and 08:26 UTC (dashed line) are compared with the total aerosol particle number concentration derived by TSMPS in the diameter range of 7 and 12 nm at 07:50 UTC, 08:10 UTC, 08:30 UTC and 08:50 UTC. All data were derived during the MelCol experiment on June 21, 2015.




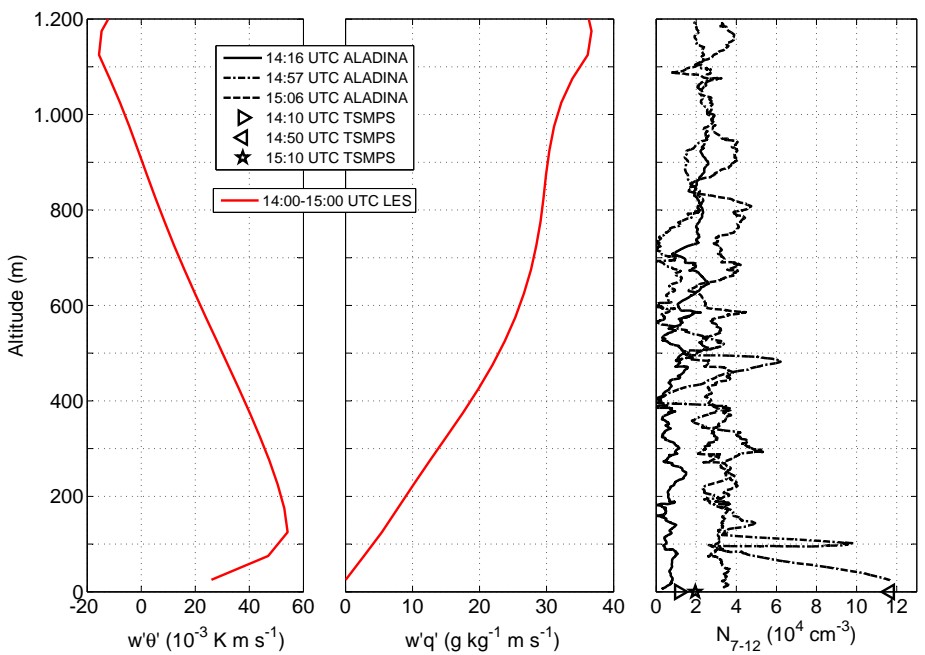

**Figure 10.** Vertical profiles of sensible heat flux $w'\theta'$ and latent heat flux $w'q'$ obtained by LES output in the interval of 14:00–15:00 UTC (red lines). Besides, three vertical profiles of freshly formed boundary aerosol in the particle diameter between 7 and 12 nm ($N_{7-12}$) measured with the CPCs on the UAS ALADINA at 14:16 UTC (solid line), 14:57 UTC (dotted line) and 15:06 UTC (dashed line) are compared with the total aerosol particle number concentration derived by TSMPS in the diameter range of 7 and 12 nm at 14:10 UTC, 14:50 UTC and 15:10 UTC. All data were derived during the MelCol experiment on June 21, 2015.





**Table 1.** Airborne measurements performed with the UAS ALADINA during Case I on 4 April, 2014. The flight time of ALADINA between take-off and landing and the maximum altitude of flights. In addition, prevailing conditions of clouds, $SO_2$ gas concentration, pressure at surface $p_0$ and temperature $T_{1m}$ from mast in 1 m height are presented in relation to the corresponding flights.

| Flight | Flight time (UTC) | Max. height (m) | Sky conditions | $SO_2$ ($\mu g\,m^{-3}$) | $p_0$ (hPa) | $T_{1m}$ (°C) |
|---|---|---|---|---|---|---|
| 1 | 06:15-06:50 | 950 | StCu | 1.7–2.4 | 998 | 7.7–8.8 |
| 2 | 07:50-08:17 | 700 | ABL clouds | 1.1–2.2 | 999 | 5.3–6.1 |
| 3 | 08:55-09:30 | 750 | ABL clouds | 1.9–2.9 | 999 | 7.0–9.1 |
| 4 | 10:41-11:28 | 1000 | ABL clouds | 3.7–4.6 | 1000 | 10.8–11.3 |
| 5 | 11:45-12:27 | 950 | ABL clouds–StCu | 2.7–4.4 | 1000 | 10.8–12.1 |
| 6 | 13:20-13:58 | 950 | StCu | 3.6–5.0 | 1001 | 11.8–12.3 |





**Table 2.** Performed airborne measurements with the UAS ALADINA during MelCol experiment Case II on June 21, 2015. The flight time of ALADINA between take-off and landing and the maximum altitude of flights. In addition, prevailing conditions of clouds, $SO_2$ gas concentration, pressure at surface $p_0$ and temperature $T_{1m}$ from mast in 1 m height are presented in relation to corresponding flight time series.

| Flight | Flight time (UTC) | Max. height (m) | Sky conditions | $SO_2$ ($\mu g\,m^{-3}$) | $p_0$ (hPa) | $T_{1m}$ (°C) |
|--------|-------------------|-----------------|----------------|--------------------------|-------------|---------------|
| 1 | 08:00-08:35 | 950 | ABL clouds–StCu | 1.2–1.6 | 1007 | 14.8–15.9 |
| 2 | 08:46-09:15 | 950 | StCu | 0.9–1.2 | 1007 | 15.4–17.4 |
| 3 | 09:28-10:01 | 950 | StCu | 0.8–0.9 | 1007 | 14.7–17.4 |
| 4 | 12:03-12:38 | 1100 | StCu | 1.2–1.5 | 1007 | 15.7–17.1 |
| 5 | 12:51-13:15 | 1200 | StCu | 1.2–1.3 | 1006 | 15.7–16.3 |
| 6 | 13:51-14:23 | 1200 | StCu | 1.4–1.5 | 1005 | 15.6–18.3 |
| 7 | 14:57-15:32 | 1200 | StCu | 1.4–3.7 | 1005 | 16.9–17.7 |