# Peer review of "Airborne observations of newly formed boundary layer aerosol particles under cloudy conditions"

_Atmospheric Chemistry and Physics, 2017_

## Referee Comment (RC1) · Anonymous Referee #1 · 19 Jan 2018

The paper by Altstädter et al. aims at showing evidence for the occurrence of NPF aloft under cloudy conditions and vertical transport of the newly formed particles close to the ground. Observations were performed from the Melpitz ground based station as well as from the unmanned aerial system ALADINA. The combination of these two datasets is of great interest, however the choice of the two case studies included in this work is not fully justified (6 cases available in total), and is even more questionable that instrument failures occurred during both of them. With the exception of some sentences which are unclear, the paper is well written. However, measurements/observations are often under-used, or used to support conclusions which sometimes do not seem to be correct, or inconsistent between the different sections. Also, references are lacking, which makes is difficult to evaluate the significance of this work compared to earlier studies; additional references would also help better understanding some of the tools/methods which are only very briefly introduced. All in all, I would not recommend the publication of the manuscript in its current form. However, because I think the dataset is highly valuable, and even if major revisions are needed, I really encourage the authors to consider submitting a new version of their manuscript. Here are some comments/suggestions to help in this process.

**Specific comments**

Comment 1, P1, L12: The use of « closure » is misleading as LES -model was used in the absence of measurements, and not to confirm observations.

Comment 2, P2, L9-12: I assume that the authors refer to the Class III events identified by Gröss et al. (2015) to support the occurrence of NPF under cloudy conditions. I think this reference should be used with caution in this context, as NPF event classification is based on a new approach in the abovementioned study. Thus, I would recommend to at least explicitly mention the use of this alternative method, especially because according to the authors those Class III events are "very weak events with very small-scale particle bursts that do not evolve into a fully developed and spatially distributed nucleation event. In any case, this class of observations includes what most researchers would call "non-events". Also, in its current form the sentence on L9-12 sounds weird to me; I would either remove the last part, starting from "and these", or replace "is possible" by "might be possible".

Comment 3, P2, L19-21: The concept of "breaking waves" should be briefly recalled/clarified. Also, I assume there is a word missing in the last part of the sentence.

Comment 4, P3, L12: The expression "the appearance of nucleation and particle growth" is confusing, and should be replaced by something like "the appearance of nucleation mode particles and their subsequent growth", as the nucleation process itself is not observed at ground level.

Comment 5, P3, L16: Regarding the expression "In contrast to typical NPF events at ground by high incoming solar irradiance". It is in my view too simplistic and a bit confusing, as it suggests that only radiation is driving the occurrence of NPF events observed close to the ground. Some other factors such as the presence of primary precursors (e.g.  $SO_2$ ) as well as the strength of their sink are also determining the process.

Comment 6, P4, L1-6: Was the effect of pressure on the CPCs detection efficiency accounted for in the data analysis or did you assume it could be neglected over the range of altitudes discussed in the present study?

Comment 7, P4, L27: I would suggest to move the reference to Manninen et al. (2010) to the next paragraph, where other NPF related results are reported, or find a way to "group" all the NPF related observations in section 2.2 in order to clarify the message.

Comment 8, P5, L10-12: More explanations should be reported about LIDAR and Ceilometer measurements/data analysis since they provide information on the occurrence of clouds, which are to some extent in the scope of the present work. At least, earlier studies involving these instruments should

be mentioned. In particular, the meaning of « range corrected signal » (captions of Fig. 3 and 5) should be explained.

Comment 9, P6, about the "Results and discussion section": It is surprising to have results about the frequency of occurrence of the phenomenon of interest in this study (i.e. occurrence of NPF in so-called non-favourable conditions) reported in the abstract (P1, L6-8) and in the conclusion (P10, L32 – P11 L2) which are not further discussed in the abovementioned section, or not even recalled. I would strongly recommend to introduce and discuss those at the beginning of section 3, before the case studies. In particular, the frequency of this process occurring close to the inversion layer should be for instance compared to that of "clear" banana shape NPF events detected at the ground level. This is needed to assess the importance of the observed phenomenon. Also, the choice of the two cases investigated in the present work is not clear, and should be supported by additional explanations, especially because data were missing from ALADINA during the selected days (CPC data during first case, meteorological parameters during second case). In specific, are those two cases representative of all 6 days when similar phenomenon was observed, or do they all display contrasting conditions?

Comment 10, P6, about "Section 3.1, Case I April 2014": In my view, the data shown in this section do not fully support the conclusions provided on P8, L10-19. In particular, I believe that some of the observations meant to evidence the transport of small particles downwards do not support the occurrence of such phenomenon. More detailed comments regarding this section are provided below.

Starting with ALDINA vertical profiles:

- Detection of small particles close to the inversion layer and/or in the vicinity of clouds has already been observed. Therefore, the authors should refer to earlier studies in order to highlight how their results are similar or differ from observations reported in the literature;
- As mentioned previously, it is complex to really "follow", and thus validate, the transport of particles downwards because of missing CPC data during 2 of the vertical profiles performed with ALADINA. Looking at the profile from 11:47 on Fig.4, high N5 values are observed in a layer between 100 and 250 m a.g.l., which seems to be disconnected from the layer above 600 m a.g.l. where high N5 values are also detected. The authors suggest that the small particles (N5-10) observed close to the ground at 13:50 were transported from the upper layer (>600 m): how can they exclude the formation of small particles in the lower layer (100-250 m), and further transport of these specific particles close to the ground?
- Also, particles in the range 5-10 nm are observed at ground level with the CPCs onboard ALADINA at 13:50, while particle number size distributions measured at the same time with the SMPS do not show any clear change in this size range compared to the distributions measured earlier during the day. How do the authors explain this? It might be useful to show the values of N5-10 calculated from SMPS measurements, in a similar way as done in Fig. 9 and 10. The analysis of fluxes such as those discussed in the second case study might also give further insight into the processes observed from ALADINA;
- Last, have the authors evaluated the threshold of  $N_{5-10}$  above which the concentrations can be considered significant? Looking at the last profile (13:50)  $N_{10} > N_5$  are observed above 800 m, and the magnitude of the resulting negative concentrations appears to be similar to the magnitude of some of the positive concentrations which are discussed in this work.

Now focussing more on ground based measurements; they do not seem to support at all the conclusions on P8, L10-12 ("the lifted layer of freshly formed particles was transported downwards, which can be further seen by the temporal appearance of the small particles of a few nm in diameter in the aerosol data at ground level"):

- Two of the small particles concentration peaks in the SMPS were observed before the occurrence of the cloud. This suggest that those were not formed under cloudy conditions (as

suggested by the title), and since they were seen well before the first vertical profile was obtained from ALADINA, they cannot really support/validate the observations from these profiles;

- Assuming the last two peaks observed during daytime might result from transport of particles downwards, why are they so sporadic? Which process (P8, L12 "other processes") could explain such a behaviour? The authors should discuss more the disappearance of these particles in the spectrum;
- It is also surprising that the authors did not discuss at all the fact that these peaks coincided almost perfectly with the peaks of NOx. This observation suggests a local source of anthropogenic particles; this is further supported by the occurrence of two of the peaks during the night, when NPF is usually not favoured.

Comment 11, P8, L15-19: The authors suggest that small particles observed close to the ground are originally formed close to the inversion layer, where, based on Fig. 4, the concentrations of large particles are significantly decreased compared to lower altitudes (and this idea is further mentioned in the conclusions, on P11, L34). This behaviour is thus not similar to what was reported by Rose et al. (2015), who found that in contrast NPF was occurring under "high" CS values, at least when compared to non-event days. Same comment applies to P11, L5-8.

Comment 12, P8, regarding instrumental setup and data analysis: Why are the instruments and methods applied different between the two case studies (eg: ceilometer/LIDAR, NOx measurement/no NOx measurement, no fluxes investigation/fluxes investigation)? I understand that LES-model was used in the second case in the absence of measurements from ALADINA. However, it would have been interesting to have similar simulations for the first case in order to evaluate the ability of the model to reproduce measurements, and give further insight into the reliability of the model outputs used in the analysis of the second case. Also, more information is needed regarding measurement of CO2 fluxes and LWP, as well as additional description/explanation for Fig. 7a. Last, the authors should provide more information about the use of this fluxes analysis in the frame of studies dedicated to NPF: to which extent is this approach novel, was it used previously, was it modified compared to earlier work?

Comment 13, P9 and following: the authors refer to their observations as nucleation; this is not correct since only >7 nm particles are observed. The use of "formation events" (eg P9 L6, L9) is also questionable, and an expression such as "the appearance of small particles" would in my view better describes the observations.

Comment 14, P9, L5-7: Based on the surface plot of the particle number size distribution (Fig. 6a), the increase of the particle concentration seems to be seen up to 30 nm (instead of 20 nm), and it is thus not correct to say that these events coincided with a decrease of the particle concentration in the range 20-50 nm.

Comment 15, P9, L22-23: The sentence "The water vapour mixing ratio q increased during the day in the vertical distribution between 0 and 1500 m" is confusing and should be rephrased.

Comment 16, P9, Figure 8: "The vertical profile of turbulent kinetic energy... showed a strong connection with the structure of the ABL" (L24-25). What does this sentence mean? Also, regarding the analysis provided on P10, L6-8: it is quite complex to discuss/assess the effect of the parameters shown on Fig. 8 a and b since the vertical distribution of  $N_{7-12}$  is shown to vary faster than the time resolution of the model outputs.

Comment 17, P10, L12: The expression "validate" might be too strong, as some of the profiles do not start from ground level, and thus do not allow for a direct comparison. This analysis is anyway very interesting, and, as mentioned previously, I would recommend to do it also for the first case study.

Comment 18, P11, L3-20: It seems that some of the conclusions regarding the first case study reported in this section are different from those provided in section 3.1. In particular, the sentence on P11 L13-14 "however ground observations did not catch the newly formed boundary layer aerosol" does not suggest anymore the transport of particles downwards, and contrasts with the sentence from P8 L10-12 "the lifted layer of freshly formed particles was transported downwards, which can be further seen by the temporal appearance of the small particles of a few nm in diameter in the aerosol data at ground level". Surprisingly, a possible connection with anthropogenic sources of particles is mentioned but NOx measurements shown in Fig 2b are not used to support this hypothesis. Also, the reference to Bianchi et al. (2016) should be reconsidered; the end of the sentence should be rephrased and the message clarified, as Bianchi et al. (2016) suggest that organic compounds from anthropogenic origin (and not CO) are involved in the NPF events they observe. CO is only used as a tracer for the anthropogenic origin/signature of the sampled air masses.

Comment 19, P11, L21-33: While the authors consider the possibility for the particles formed aloft to have disappeared on L24, they assert on L25-27 that these particles are those observed close to the ground after vertical mixing. These sentences seem to be a bit contradictory, and the authors should be more careful in providing their conclusions, and use some formulations such as "might", "could", "suggest" ... In particular, the analysis of the particles and CO2 fluxes used to support the conclusions seems to be uncomplete. Indeed, while CO2 fluxes show similar behaviour during the events observed at 10:10 and 16:10, particles fluxes show contrasting behaviour, being slightly positive during the first event and negative during the second one. This cannot be simply summarized as "During sporadic formation events, significant deposition occurred, taken from the negative values of particle fluxes" (P9, L13-15). Also, as already suggested for the first case study, the authors should discuss more the disappearance of the particles.

Comment 20, P11, L23: "The maximum ... and increased rapidly while descending". This sentence is confusing, please try to rephrase.

**Technical corrections**

P3, L14: "high" should be replaced by "large".

P5, L10: "and" instead of "und".

P6, and following sections: For clarity I would suggest to give all particle concentrations as N, and not dN/dlog(dp).

P6, L8: "of" instead of "on".

P7, L11-12: I would suggest to change the expression "in the particle diameter". The authors can simply refer to N5, N10, N5-10, N390, N500 and N700 as those were defined before.

P7, L31: Please try to reformulate the expression "particles belonging to the diameter".

P8, L32: I would suggest to change the expression "in the particle diameter".

P9, L1: "in the particle diameter range between 3 and 10 nm" instead of "of 3 and 10 nm".

P10, L6: "790 m" instead of "790 nm".

P10, L7: "N7-12 decreased to the total aerosol number concentration...". This is confusing, please try to reformulate.

P10, L16: "integration of a size scan".

P11, L7: "high concentrations of sulphuric acid".

P17, Fig. 1: I would suggest to show the location of the station on the map, even if it said that the station is located in the domain centre.

P18, Fig. 2: the ticks on the x axis do not ease the reading of the times (10 intervals for 6 hours). Same applies to Fig. 6.

P20, Fig 4: Logarithmic scales should be used for particle concentrations as it is very difficult to seen the variations of N5-10 and N700.

---

## Referee Comment (RC2) · Anonymous Referee #2 · 27 Jan 2018

In summary this paper presents observations of non-surface layer new particle formation under cloudy conditions and near surface vertical transport of newly formed particles. Observations in the vertical were made using the UAS system ALADINA in the region of the TROPOS research site near Melpitz: TROPOS facilities providing surface observations. 53 hours of flight data was available spanning a period 23 days between October 2013 and July 2015. Of the available data two periods were selected as case studies.

Although the paper is well structured and presented the quality of the English is poor making for a difficult read and confusion as to what point the author is trying to convey

[Figure]

– there are far too many corrections to list here. There is also a tendency for the author to uses emotive\subjective phrases for example: "All in all, the research site Melpitz of TROPOS offers a great potential for observing NPF within the ABL and inter-comparison of airborne data with monitoring at ground." These are major failings and ones that need to be addressed: if the author is not a native English speaker it is recommended that they find somebody who is to assist in editing.

The work presented is unique and will be of great interest to the community but there needs to be a substantial "tightening up". The paper gives the impression of slackness and carelessness in experimental techniques, discussion of relevant processes, and how conclusions have be arrived at. At this stage and in this form I would not recommend publication but I would encourage the authors to re-submit after revision.

Some suggested areas to revisit. 1. The authors site a number of papers concerning the experimental set up at both the field site on for the UAS. It would be expected for a basic discussion of both these setups to be included in the paper and a discussion of the various factors that will effect measurement uncertainty and what has been done to mitigate them. This is very important when it comes the particulate measurement as inlet loses and transmission losses can seriously bias a measurement: for example is the fact that you are not seeing particle above a certain size because they are not there in the first place or because they have all been lost before reaching the sensors. Basically have you characterised the both the ground and UAS sampling and measuring systems. If you have then discuss it and the implications. If this has not been done you would be advised to do so.

2. There is also the point regarding humidity and this applies to both the ground and UAS measurements. You are measuring the humidity of the ambient air sample upstream of your sensors. If that air mass is warmed (resulting in its RH reducing) or purposefully dried then this will impact on particle size. There is a need to be exact as to what the conditions are at the point of measurement and to indicate what effect this might have.

3. The need to be exact also extends to terms you are using. You refer to size but what do you actually mean: radius, diameter, and are these optical, mobility or aerodynamic equivalent values. This needed to be exact also extends to the gas phase measurements: are the concentration in ambient air or dry air? What are the uncertainties in these measurements?

4. With regard to your CPCs. In the first instance you are using two units: one with a lower cut of 5nm (diameter?) and one with a lower cut off of 10nm (diameter?) while in the second you are using two units: one with a lower cut off of 7nm (diameter?) and one with a lower cut off of 12nm (diameter?) and on page 4 lines 1 – 6 this is clear. Lines 10 – 14 on page 7 this is not the impression the text gives. This is what you say: "total aerosol particle number concentration measured with two CPCs in the particle diameter of 5 nm (red line) and 10 nm (blue line),". What I think you are talking about is the total aerosol concentration measured by the two cpc: red line cpc with 5nm lower cut off, blue line the cpc with a 10nm lower cut off. On line 12 of the same page you say: "OPC in the particle diameter of 390 nm (pink line), 500 nm (green line) and 700 nm (turquoise line)". An OPC has specific measurement bins that will have a centre value of particle diameter and a specified width – you need to add the width of the bin to this kind of statement and state explicitly that you are using the bin centre value.

5. Your figures show no error bars nor is there any reference in the captions: this needs to be addressed. On the subject of figures, although I am not colour blind many of your readers may be. It is good practice to choose colour schemes that those who are colour blind can interpret.

6. In your discussion of the use of the LES you use the term "closure": however what you are actually doing is using the model output to substitute for missing observations. Again the need for exactness but there is also the question of the validity of this approach. What evidence can you present that would indicate that the LES has accurately representing the conditions at the site. Presenting a comparison the model data with flights where you have full observations would be advised.

7. Lidar and ceilometer. You put a deal of emphasis on the importance of these observations without the necessary explanation and discussion of the observation themselves. What are you actually looking at what is the implication to the interpretation arising from the change of laser wavelength?

8. In your paper introduction you introduce many concepts and make many statements backed up by a paper reference – you could really do with expanding this by adding brief textural explanations.

9. You introduce "gravity waves" but do not provide an explanation as to why this is relevant and the process that are at play.

10. Use of the term inter-comparison. This is a tautology and a very common mistake – simply use the term comparison.

---

## Short Comment (SC1) · 19 Feb 2018

The paper shows some interesting data on elevated ultrafine particle layers under total overcast conditions obtained, using an unmanned aerial vehicle, the UAV ALADINA, flying vertical profiles through the planetary boundary layer and lower free troposphere. The occurrence of ultrafine particles under such meteorological conditions would not be expected following the general description of new particle formation observed elsewhere. The authors nevertheless, claim a clear identification of a new particle formation event. Typically for the production of ultrafine particles via new particle formation or, with the old expression, gas-to-particle conversion, shortwave radiation is suggested to

be required as an initial step to convert first sulphur dioxide into sulphuric acid, which than reacts in the atmosphere often with ammonia or similar compounds producing initial aerosol clusters. This is well established in a large number of publications, see for example for a review Kulmala et al, 2011, and Kulmala et al, 2013 for an analysis of the expected particle generation process. Growth occurs then via (EL)VOC, further sulphuric acid water etc. . The data shown here were measured under conditions without sufficiently high UV radiation levels to produce sulphuric acid in the atmosphere, thus requiring a different production process. Such an occurrence of ultrafine particles under totally overcast conditions has been observed several times and was reported for example over Germany (Junkermann et al, 2016), a paper that is cited in the manuscript. Here these particles were apportioned to primary industrial sources burning fossil fuel. Hence, as several of these elevated sources are located around Melpitz the observations are not really surprising. Anyway, the data analysis leaves several questions

1) The manuscript states that NPF occurrence with subsequent downward transport was clearly identified.

The question: What does 'new' mean within this context? Was the particle formation (cluster production and subsequent growth) really been observed?

How can a nucleation (gas to particle conversion) be identified with instrumentation that is not able to measure the initially formed particles? From the two size fractions the size of the major nucleation or Aitken mode cannot be estimated. It could be anything larger than 12 nm and only shows, that the distribution contains some particles in the range below 10 or 12 nm.

A gas to particle (NPF) event however, should have as a minimum requirement initially the majority of particles in the lowest size bins of the distribution. The number of particles in the size between 7 and 12 nm (or N7-12) has then to be higher than the number of particles > 12 nm before and at the time the growing nucleation mode is
passing this size range. Here, within the data presented the difference between the size bins is marginal, compared to the overall number concentrations indicating rather an aged air mass. Occasionally the N10 is even higher (measurement uncertainty?) than N5., in Fig. 4b and 4c N10 is missing at all.

2) Following Kulmala et al, 2013, it takes several,  $\sim$  about five or more, hours for a new particle to grow into the lowest size range of the particles that can be observed with the instruments on ALADINA. Observation of particles at 08:00 UTC thus requires a production at $\sim$  03:00 UTC or even earlier during the night. These particles have to be generated in a different area, a minimum of five hours upwind, which is according to HYSPLIT at least 70 km. Growth rates are normally smaller or even zero at night. Thus the source could be even further away.

3) Where is the production area of the particles, which are 'clearly' identified as originating from NPF, according to HYSPLIT  $\sim$  70-100 km for 5 h?

4) Why are the meteorological and atmospheric chemical conditions favorable for new particle formation > 70-100 km upwind, during the night and in the elevated layer? Where do the precursor chemical compounds come from? What could be the initial step for nucleation cluster generation at night?

5) Is there probably a source for ultrafine particles upwind that could explain the results? HYSPLIT shows that the air mass has been close to the German-Polish border on the evening of April 3. Here we have at least three power stations that are sources for primary particles in the respective size ranges as well as additional large amounts of sulphur dioxide, ammonia (from the SCR cleaning process) and also internally produced sulphuric acid (see also Junkermann et al, 2016). Size distributions, independence on time of the day (Junkermann and Hacker, 2015) and laboratory results confirm primary emission of 'New' particles (Brachert et al, 2013). Such particle production does not require UV radiation or OH radicals.

A quick HYSPLIT analysis: For June 21 the winds in the altitude of 600 m above
ground come from the northwest passing the power station Buschhaus as the next possible candidate. Further upwind (up to 24 h) along the backtrajectory the industries of Groningen (NL) are located. For westerly to southwesterly winds in the PBL the power stations Schkopau and Lippendorf as well as the refinery Leuna would be possible sources for primary ultrafine particles and further precursors over Melpitz (see also Platis et at, 2015). For size distributions and source strength of such sources see Junkermann et al, 2011; 2016.

6) Page 9, line 20, atmospheric boundary layer conditions were unfortunately not available from the UAS. Instead a model is used to characterize the ABL. However, this model is not able to reproduce the measured vertical structure (Fig. 9). Why are no model data shown for 07:00 to 08:00, the time window before the aircraft measured the pronounced vertical profile? Lateron the PBL rapidly mixed. The vertical structure was visible only within the first 15 minutes of the model 1 h time window.

7) Fine particles: particle number concentrations > 390 nm on April 4 are fairly high. We can compare to particle numbers measured with a GRIMM 1.108 (fraction > 400 nm, second size bin) over Kathmandu (missed approach to Kathmandu international airport) in January 2014, to data gathered during flights in the extremely dusty Mexico City basin during Milagro 2006 or to data from the heavily polluted Po-Valley (QUEST, 2004). The number concentrations there were only about half or less of what has been seen over Melpitz. Under these conditions the condensational sink (CS) would be extremely high and according to most of the published literature a NPF event would be very unlikely. How is this in agreement with the summary / abstract statement that New Particle Formation has been clearly observed?

The summary claims: Further, the NPF event was 5 linked to a condensation sink of larger particles belonging to the accumulation mode at the same altitude. That's not in agreement with the fig. 4, the layers are clearly separated.

8) Fig. 9 and page 11, line 22/23: 'The maximum of total aerosol particle number
concentration was 1.6 $\times105$  cm–3 at the altitude of 420ma.g.l. and increased rapidly while descending'.

Not in agreement with the figure. The number concentration below 420 m actually is always lower than the maximum at 420 m and decreased between 08:13 and 08:26 UTC. Probably also a result of changing wind directions ( $\sim$  15 deg within 2 hours)

Summarizing: The paper does not really show evidence or proof for gas to particle conversion respective new particle formation. Though interesting and good to identify ultrafine particle layers the instrumentation used is most probably not appropriate to detect such particle formation events. An SMPS or NAIS would be necessary.

The results are contrary to the general literature about new particle formation and unfortunately, there is no attempt to analyze where the particles observed are produced or originating from. However, the measurements would be in agreement with emissions from a few welll known 'continuous generators for ultrafine particles' e.g. power stations /refineries and the final statement that a thorough meteorological analysis (but, not only in the vertical) is necessary to understand ultrafine particle behavior in the PBL is strongly supported.

References:

Brachert, L., Kochenburger, T., and Schaber, K., 2013: Facing the Sulfuric Acid Aerosol Problem in Flue Gas Cleaning: Pilot Plant Experiments and Simulation, Aerosol Science and Technology, 47, 1083–1091

Junkermann, W., Vogel, B. and Sutton, M.A. 2011: The climate penalty for clean fossil fuel combustion, Atmos. Chem. Phys., 11, 12917-12924

Junkermann, W., and Hacker, J.M., 2015: Ultrafine particles over Eastern Australia: an airborne survey, Tellus B, 67, 25308, http://dx.doi.org/10.3402/tellusb.v67.25308

Junkermann, W., Vogel, B. and Bangert, M., 2016: Ultrafine particles over Germany - an aerial survey, Tellus B, 2016, 68, 29250, http://dx.doi.org/10.3402/tellusb.v68.29250

**ACPD**
Kulmala, M., and Coauthors, 2011: General overview: European Integrated project on Aerosol Cloud Climate and Air Quality interactions (EUCAARI) – integrating aerosol research from nano to global scales, Atmos. Chem. Phys., 11, 13061-13143, https://doi.org/10.5194/acp-11-13061-2011, 2011

Kulmala, M. and Coauthors, 2013: Direct Observations of Atmospheric Aerosol Nucleation, Science, 339, 943-946, doi: 0.1126/science.1227385 Platis, A., and Coauthors, 2015: An Observational Case Study on the Influence of Atmospheric Boundary-Layer Dynamics on New Particle Formation, Boundary-Layer Meteorology, DOI 10.1007/s10546-015-0084-y

**ACPD**

---

## Author Comment (AC1) · 25 Apr 2018

**Authors' response and marked-up manuscript**

**Manuscript number: acp-2017-1133**

**Airborne observations of newly formed boundary layer aerosol particles under cloudy conditions by Altstädter et al.**

*First of all, the authors would like to thank both referees and Wolfgang Junkermann for their valuable support and constructive criticism that will improve the present form of the manuscript. Just to make clear the choice for the selected days on April 4, 2014 and June 21, 2015: The two days are of special interest in terms of similar weather conditions (strato cumulus clouds) influenced by a passage of low pressure system and low precursor gases (of SO2), but affected by different wind directions in order to investigate the impact of anthropogenic emissions on new particle formation, as assumed by e.g. Junkermann et al. (2016). During our observations, however, no clear new particle formation event with high and continuous growth rate was observed at ground. Only sporadic clusters were formed during short intervals of cloud gaps. NPF formation between clouds has been recently studied by Wehner et al. (2015), but the study differs from our observations, due to (now) high turbulence on local scale. In our opinion, the study of the goal is now clarified, supported by your contributions. Thank you!*

*On April 4, the site was influenced by east wind direction, so that anthropogenic emissions due to polluted air masses from Eastern Germany might have been observed that coincide with the increase of NOx and will confirm the study of Junkermann et al. (2016). At this point, we do not exclude the impact of anthropogenic emissions on NPF, but in our opinion, the sources are local and not from long distances. With the help of the UAS profiles, an increase of $N_{5-10}$ was observed in connection to a condensation sink, shown by OPC data, and enhanced moisture above the inversion layer at around 9 UTC, which spread out even further until noon around 14 UTC.*

*The overall observation was that the new particle formation event was not captured by the TSMPS at ground, but was proven by the UAS (even in an altered state). Similar sporadic appearances of ultrafine particles were investigated on June 21. During the second measurement day, the profiles were performed as part of the MelCol2015 campaign. Therefore, additional instrumentation was available at the same time. We used data from an eddy-covariance station that confirmed the rapid upward and downward transport of aerosol particles, so that mixing processes, initiated by large turbulent fluxes within the ABL, led to insufficient particle growth rate. Turbulent fluxes were estimated by an LES model that was used for the campaign. Unfortunately, there was no meteorological data available, therefore the ABL was characterised by the LES. In our opinion, the second measurement day is still interesting and unique by the use of various instrumentations that further link to UAS profiles and ground observations.*

*As the whole set up was presented in previous studies already (Altstädter et al., 2015; Platis et al., 2016 and Bärfuss et al., 2018), the authors decided "just" to briefly introduce the used instrumentation. In our opinion, the aerosol and meteorological sensors are sufficiently described in Section 2.1 (information on manufacturer, accuracy and response time). Nevertheless, we added the following lines on p.4, l. 6-9:*

The aircraft design and instrumentation's set up were introduced in Altstädter et al. (2015) and supported by the results of Platis et al. (2016). During review process, the aircraft was re-engineered and we refer to the newest study of Bärfuss et al. (2018) who presented a profound overview of ALADINA's flight operation, data procedure and the current payload that" is equipped with aerosol instrumentation and meteorological sensors with high temporal resolution.

*(…) p4., 16-18:*

The laboratory results were confirmed by Bärfuss et al. (2018) during field studies in Melpitz. The UAS was placed at the same aerosol inlet as ground monitoring and N7-12 coincide with ground data in the same particle size within the deviations of +/+20% during sampling period of 1.5 h.

*Please, find attached the authors' responses and the marked-up manuscript including the changes and remarks corresponding to the referees in different colours (RC1 in green, RC2 in blue, SC1 in yellow), respectively.*
*In addition, during review process, one of the co-authors has changed the affiliation and some references were revised. Please, take into account the remarks given by the authors at the end of these comments (see 4th point, marked-up in magenta).*

**Comments of referee #1 (RC1)**

"The paper by Altstädter et al. aims at showing evidence for the occurrence of NPF aloft under cloudy conditions and vertical transport of the newly formed particles close to the ground. Observations were performed from the Melpitz ground based station as well as from the unmanned aerial system ALADINA. The combination of these two datasets is of great interest, however the choice of the two case studies included in this work is not fully justified (6 cases available in total), and is even more questionable that instrument failures occurred during both of them. With the exception of some sentences which are unclear, the paper is well written. However, measurements/observations are often under-used, or used to support conclusions which sometimes do not seem to be correct, or inconsistent between the different sections. Also, references are lacking, which makes is difficult to evaluate the significance of this work compared to earlier studies; additional references would also help better understanding some of the tools/methods which are only very briefly introduced. All in all, I would not recommend the publication of the manuscript in its current form. However, because I think the dataset is highly valuable, and even if major revisions are needed, I really encourage the authors to consider submitting a new version of their manuscript. Here are some comments/suggestions to help in this process."

**Author's response to referee #1**

*The authors acknowledge the anonymous referee for the detailed proof reading and for the specific suggestions that will highlight the outcome of the current manuscript. The authors agree that some parts might irritate the reader, as the additional instrumentation was not the same during the two measurement days. Please, take into account the remarks on p. 1 of the authors' response.*

*In the following, the authors will go through the comments that were given by the first referee.*

*Please, find attached the point-to point response and the marked-up manuscript version including the changes (removed is marked in red and new is coloured in green).*

**Point-to-point response:**

**RC1**: "Comment 1, P1, L12: The use of « closure » is misleading as LES -model was used in the absence of measurements, and not to confirm observations."

*Authors' response: The authors agree that the sentence contains a false statement, as the LES model was used in order to derive meteorological data due to missing data by the UAS and not for comparison.*
*We have reformulated the last sentences of the abstract in order to state clear the outcome of the study on p. 1, l. 16-20:*

*During the first measurement day, a maximum of $N_{5-10}$ =3.5x10^3 cm^-3 was detected above an inversion layer in connection with a decline of larger particles and layer of high moisture that existed until noon. However, altered clusters (particle size between 30 and 50 nm) were measured at ground during the process in relation to an increase of NOx so that anthropogenic emissions might have been favoured the NPF occurrence. Whereas by south-west wind conditions, pronounced turbulent fluxes of sensible and latent heat, taken from an LES model output" (..)*

**RC1**: "Comment 2, P2, L9-12: I assume that the authors refer to the Class III events identified by Gröss et al. (2015) to support the occurrence of NPF under cloudy conditions. I think this reference should be used with caution in this context, as NPF event classification is based on a new approach in the above mentioned study. Thus, I would recommend to at least explicitly mention the use of this alternative method, especially because according to the authors those Class III events are "very weak events with very small-scale particle bursts that do not evolve into a fully developed and spatially distributed nucleation event. In any case, this class of observations includes what most researchers would call "nonevents". Also, in its current form the sentence on L9-12 sounds weird to me; I would either remove the last part, starting from "and these", or replace "is possible" by "might be possible"."

*Authors' response: Thank you very much for that hint. The authors agree that the comparison with Class III events presented in Größ et al. (2015) (now: Größ et al (2018)) is irritating. According to the revised version of the manuscript, the authors have adjusted their descriptions. Class III events were classified as "low NPF intensity, including non-events". Similar observations during clouds were measured with NAIS called case 4, Fig. 3d. The literature was removed to Section 3.2, because the authors think that this reference is similar and needs to be addressed (p. 10, l. 12-15):*
*A similar observation of clustered NPF during cloudy conditions and west wind in Melpitz was shown in the recent study of Größ et al. (2018). Insufficient particle growth rate was significantly connected to fluctuations of hydroxyl radicals and H2SO4 (shown in the so called Case 4) that were steadily suppressed by clouds in contrast to high increase of H2SO4 during clear sky. The authors suggested local processes and steadily influenced sources as cause for this formation behaviour.*

*As the authors wanted to stress the hypothesis that nucleation is expected more often than assumed so far in current literature, the authors changed the lines "is possible" into "might be possible" (see p.2, l. 16).*

**RC1**: "Comment 3, P2, L19-21: The concept of "breaking waves" should be briefly recalled/clarified. Also, I assume there is a word missing in the last part of the sentence."

*Authors' response: Thank you a lot for the correction. Yes, the sentence is not complete in the present form. Further we believe that the mentioning of breaking waves is in fact not relevant for the scope of the paper and we decided to skip that in the modified version of our manuscript (p. 2, 26-28).*

*Further, Bigg (1997) suggested that high humidity and temperature fluctuations may enhance new particle formation. Besides, Easter and Peters (1994) (…)*

**RC1**: "Comment 4, P3, L12: The expression "the appearance of nucleation and particle growth" is confusing, and should be replaced by something like "the appearance of nucleation mode particles and their subsequent growth", as the nucleation process itself is not observed at ground level."

*Authors' response: The authors acknowledge the first referee for the comment. The suggested lines "the appearance of nucleation mode particles and their subsequent growth" were changed into the marked up manuscript on p.3, l.19-20.*

**RC1**: "Comment 5, P3, L16: Regarding the expression "In contrast to typical NPF events at ground by high incoming solar irradiance". It is in my view too simplistic and a bit confusing, as it suggests that only radiation is driving the occurrence of NPF events observed close to the ground. Some other factors such as the presence of primary precursors (e.g. SO2) as well as the strength of their sink are also determining the process."

*Authors' response: The authors intend to highlight the observations during cloudy conditions in contrast to the previous results by clear sky conditions, shown in Platis et al. (2016). Indeed, further factors cause new particle formation, as mentioned in previous paragraph (see p. 2, l. 8-10 of the discussion paper). But in this case, the authors stated "typical NPF events at ground by high incoming solar irradiance" as the main difference from previous studies to the current manuscript. In order to avoid any lacking description, the authors reformulated the sentences (p. 3, l. 15-21):*
*Previous investigations of Platis et al. (2016) showed NPF in correlation with temperature and humidity fluctuations of several orders of magnitudes higher than in the remaining part of the ABL. In addition, downwards transport of freshly formed particles to ground level was observed and supported by the appearance of nucleation mode particles and their subsequent growth relating to an increase of $SO_2$ concentrations by clear sky.*

**RC1**: "Comment 6, P4, L1-6: Was the effect of pressure on the CPCs detection efficiency accounted for in the data analysis or did you assume it could be neglected over the range of altitudes discussed in the present study?"

*Authors' response: The CPC data was not corrected for pressure effect, because for the pressure range of interest, 900 -1025 hPa, the counting efficiency of the CPC changes only minimal, i.e., less than 4% (Heintzenberg et al., 1999), which is still within the range of the overall CPC uncertainty.*

**RC1**: "Comment 7, P4, L27: I would suggest to move the reference to Manninen et al. (2010) to the next paragraph, where other NPF related results are reported, or find a way to "group" all the NPF related observations in section 2.2 in order to clarify the message."

*Authors' response: The authors decided to split the section into four paragraphs in order to obtain a clear structure: 1. Short description of measurement site, 2. air mass characterisation, 3. previous studies of NPF, 4. used instrumentation in this study.*

*The transition was done by the change on p. 5, l. 9-10; in order to group the following reference of Manninen et al., 2010: The research site Melpitz of TROPOS is well known in NPF studies and benefits from comparison of airborne data with monitoring at ground.*

**RC1**: "Comment 8, P5, L10-12: More explanations should be reported about LIDAR and Ceilometer measurements/data analysis since they provide information on the occurrence of clouds, which are to some extent in the scope of the present work. At least, earlier studies involving these instruments should be mentioned. In particular, the meaning of « range corrected signal » (captions of Fig. 3 and 5) should be explained."

*Authors' response:* The authors agree that lidar and ceilometer are not sufficiently introduced or referenced. However, the authors' intention is to **show briefly** the existence of clouds and not to go any deeper into the description of the instrumentation.
The range-corrected signal is the uncalibrated attenuated backscatter coefficient. Attenuated by molecules and particles. As at 1064 nm attenuation is weak, it is a good proxy for the particle backscatter coefficient.

*Following lines and references were added (p. 5, l. 25-29):*

*Clouds and aerosol layers are indicated in the lowermost 3–4 km with backscatter signals of ceilometer (in the wavelength of 1064 nm; e.g. Heese et al., 2010; Wiegner and Geiß, 2012) and lidar (in the wavelength of 532 nm; e.g. Pal et al., 1992; Engelmann et al., 2016). During the first measurement day, ceilometer data is taken into account, whereas during the MelCol 2015 experiment, the PollyXT lidar (e.g. Althausen et al., 2009) was available. The given range corrected signal is the uncalibrated attenuated backscatter coefficient that is damped by molecules and particles, respectively.*

*- Engelmann, R., Kanitz, T., Baars, H., Heese, B., Althausen, D., Skupin, A., Wandinger, U., Komppula, M., Stachlewska, I. S., Amiridis, V., Marinou, E., Mattis, I., Linn\'{e}, H., and Ansmann, A.: The automated multiwavelength Raman polarization and water-vapor lidar PollyXT: the neXT generation, Atmos. Meas. Tech., 9, 1767-1784, https://doi.org/10.5194/amt-9-1767-2016, 2016.*
*-Heese, B., Flentje, H., Althausen, D., Ansmann, A., and Frey, S.: Ceilometer lidar comparison: backscatter coefficient retrieval and signal-to-noise ratio determination, Atmos. Meas. Tech., 3, 1763-1770, https://doi.org/10.5194/amt-3-1763-2010, 2010.*
*-Pal, S. R., W. Steinbach, and A. I. Carswell, 1992: Automated method for lidar determination of cloud-base height and vertical extent. Applied Optics,31, 1488–1494*
*-Wiegner, M. and Geiß, A.: Aerosol profiling with the Jenoptik ceilometer CHM15kx, Atmos. Meas. Tech., 5, 1953-1964, https://doi.org/10.5194/amt-5-1953-2012, 2012.*

**RC1**: "Comment 9, P6, about the "Results and discussion section"": It is surprising to have results about the frequency of occurrence of the phenomenon of interest in this study (i.e. occurrence of NPF in so-called non-favourable conditions) reported in the abstract (P1, L6-8) and in the conclusion (P10, L32 – P11 L2) which are not further discussed in the abovementioned section, or not even recalled. I would strongly recommend to introduce and discuss those at the beginning of section 3, before the case studies. In particular, the frequency of this process occurring close to the inversion layer should be for instance compared to that of "clear" banana shape NPF events detected at the ground level. This is needed to assess the importance of the observed phenomenon. Also, the choice of the two cases investigated in the present work is not clear, and should be supported by additional explanations, especially because data were missing from ALADINA during the selected days (CPC data during first case, meteorological parameters during second case). In specific, are those two cases representative of all 6 days when similar phenomenon was observed, or do they all display contrasting conditions?"

*Authors' response:* Just one comment: At the moment, another study is in preparation that focuses on statistics of ultrafine particles obtained by profiling with ALADINA. However, this is not the scope of the current paper. During our observations, only one "clear" banana shape occurred on July 1, 2015 in relation to high SO2 concentrations and updraft of ultrafine particles during clear sky. The results were presented at the European Aerosol Conference 2017 in Zurich, already. However, the whole study is still being processed, as it belongs to the MelCol 2015 campaign. The authors decided not to introduce the campaign, as an overview paper is currently in progress and will be published in the near future. Regarding the other points, we would like to refer to the very beginning of the authors' response on p. 1. In our opinion, the two case studies are representative, as the main wind directions were considered. We agree that the high occurrence of NPF phenomena during cloudy conditions should be revised in the results. Therefore, the sentences were added on p. 6, l. 25-28:
*To almost one third of the performed flight days with ALADINA, bursts of ultrafine particles were observed under cloudy conditions close to the inversion layer. Two of the six measurement days were selected explicitly due to similar weather conditions, low SO2 concentrations, but different wind (NE, SW) directions that however mainly influenced the site (e.g.Spindler et al., 2012; Engler et al., 2007).*

**RC1**: "Comment 10, P6, about "Section 3.1, Case I April 2014"": In my view, the data shown in this section do not fully support the conclusions provided on P8, L10-19. In particular, I believe that some of the observations meant to evidence the transport of small particles downwards do not support the occurrence of such phenomenon. More detailed comments regarding this section are provided below.
Starting with ALDINA vertical profiles:

- Detection of small particles close to the inversion layer and/or in the vicinity of clouds has already been observed. Therefore, the authors should refer to earlier studies in order to highlight how their results are similar or differ from observations reported in the literature;

- As mentioned previously, it is complex to really "follow", and thus validate, the transport of particles downwards because of missing CPC data during 2 of the vertical profiles performed with ALADINA. Looking at the profile from 11:47 on Fig.4, high N5 values are observed in a layer between 100 and 250 m a.g.l., which seems to be disconnected from the layer above 600 m a.g.l. where high N5 values are also detected. The authors suggest that the small particles (N5-10) observed close to the ground at 13:50 were transported from the upper layer (>600 m): how can they exclude the formation of small particles in the lower layer (100-250 m), and further transport of these specific particles close to the ground?

*Authors' response:* *Thank you for the comment. Indeed, the description is misleading. We cannot exclude a second layer with ultrafine particles in the heights between 100 and 250 m. However, we doubt that the layer is mixed from the upper part, as the capping inversion is still existent.*

- Also, particles in the range 5-10 nm are observed at ground level with the CPCs onboard ALADINA at 13:50, while particle number size distributions measured at the same time with the SMPS do not show any clear change in this size range compared to the distributions measured earlier during the day. How do the authors explain this? It might be useful to show the values of N5-10 calculated from SMPS measurements, in a similar way as done in Fig. 9 and 10. The analysis of fluxes such as those discussed in the second case study might also give further insight into the processes observed from ALADINA;

*Authors' response:* *As it was discussed in previous parts, during the two measurement days, different instrumentation were available at the TROPOS research site. We inserted one comment for the reader in order to avoid any confusion of the different instrumentation in Section 3.2. (p. 9, l. 17-18)*

*ALADINA flights were performed as part of the MelCol 2015 experiment so that additional instrumentation was available at site.*

- Last, have the authors evaluated the threshold of N5-10 above which the concentrations can be considered significant? Looking at the last profile (13:50) N10 >N5 are observed above 800 m, and the magnitude of the resulting negative concentrations appears to be similar to the magnitude of some of the positive concentrations which are discussed in this work.
Now focussing more on ground based measurements; they do not seem to support at all the conclusions on P8, L10-12 ("the lifted layer of freshly formed particles was transported downwards, which can be further seen by the temporal appearance of the small particles of a few nm in diameter in the aerosol data at ground level"):

*Authors' response:* *The authors agree that the statement is wrong. A brief summary is given later on (p.9, l. 12-25). There is no proof of downwards transport in the first case and in the following, the sentences (p.9, l. 6-9) were deleted and explanations are given now in the concluding remarks.*

- Two of the small particles concentration peaks in the SMPS were observed before the occurrence of the cloud. This suggest that those were not formed under cloudy conditions (as suggested by the title), and since they were seen well before the first vertical profile was obtained from ALADINA, they cannot really support/validate the observations from these profiles;

- Assuming the last two peaks observed during daytime might result from transport of particles downwards, why are they so sporadic? Which process (P8, L12 "other processes") could explain such a behaviour? The authors should discuss more the disappearance of these particles in the spectrum;

*Authors' response:* *In our opinion, the "other processes" and most of the questions in Comment 10 are well explained by the response to the short comment and added to the conclusion (p. 13, l. 21-31).*

*However, no clear typical formation event like a "banana shape" was identified by ground data. Bursts of ultrafine particles occurred in connection with increase of NOx concentrations affected by east wind conditions under dense strato cumulus clouds. The authors' hypothesis is that the events observed here are locally restricted and not homogeneous over larger regions. This is because in particular the clouds create inhomogeneities in the intensity of solar radiation reaching the ground. On the other hand, clouds and their surrounding regions, as well as inversion layer, are influenced by large gradients in various parameters. These gradients in combination with local turbulence may create strong nonlinearities being a favourable environment for new particle formation which has been discussed theoretically by Easter and Peters (1994) and shown in-situ by Wehner et al. (2015). The dynamics of such an event being locally restricted to small regions are completely different and cannot be explained by characteristics of a large-scale event and relation to air mass.*

*The small NPF events may occur suddenly, followed by fast growth but also fast dilution with surrounding air, seen in sporadic clusters of TSMPS data.*

- It is also surprising that the authors did not discuss at all the fact that these peaks coincided almost perfectly with the peaks of NOx. This observation suggests a local source of anthropogenic particles; this is further supported by the occurrence of two of the peaks during the night, when NPF is usually not favoured."

*Authors' response: Further, we have considered the fact, that NOx coincide with the appearance of small particles measured with the TSMPS in Figs. 2a,b (see p. 1, 17-19; p. 7., l. 17-19).*

*During the first measurement day, a maximum of $3.5 \times 10^3$ cm$^{-3}$ for $N_{510}$ was detected above an inversion layer in connection with a condensation sink and layer of high moisture that existed until noon. However, altered clusters (particle size between 30 and 50 nm) were measured at ground during the process in relation to an increase of $NO_x$ so that anthropogenic emissions might have been favoured the NPF occurrence.*

*The values of NOx varied between 10.2 and 34.6 µg m$^{-3}$ during the day, with temporary higher concentrations in the early morning at 03:50 UTC, from 07:55 to 09:40 UTC and in the afternoon around 18:00 UTC that further coincide with the appearance of ultrafine particles estimated with the TSMPS data.*

**RC1**: "Comment 11, P8, L15-19: The authors suggest that small particles observed close to the ground are originally formed close to the inversion layer, where, based on Fig. 4, the concentrations of large particles are significantly decreased compared to lower altitudes (and this idea is further mentioned in the conclusions, on P11, L34). This behaviour is thus not similar to what was reported by Rose et al.(2015), who found that in contrast NPF was occurring under "high" CS values, at least when compared to non-event days. Same comment applies to P11, L5-8."

*Authors' response: Yes, indeed. We have deleted the phrases and compared our studies with more suitable literature of Easter and Peters (1994) and Wehner et al. (2015) instead of Rose et al. (2015). For detailed explanations, please take into account the answers to the short comment.*

**RC1**: "Comment 12, P8, regarding instrumental setup and data analysis: Why are the instruments and methods applied different between the two case studies (eg: ceilometer/LIDAR, NOx measurement/no NOx measurement, no fluxes investigation/fluxes investigation)? I understand that LES-model was used in the second case in the absence of measurements from ALADINA. However, it would have been interesting to have similar simulations for the first case in order to evaluate the ability of the model to reproduce measurements, and give further insight into the reliability of the model outputs used in the analysis of the second case. Also, more information is needed regarding measurement of CO2 fluxes and LWP, as well as additional description/explanation for Fig. 7a. Last, the authors should provide more information about the use of this fluxes analysis in the frame of studies dedicated to NPF: to which extent is this approach novel, was it used previously, was it modified compared to earlier work?"

*Authors' response: As it was stated already, the LES model and EC station were only available during the second measurement day, as part of the MelCol 2015 campaign. At this point, we point to the detailed explanations on p.1 of the authors' response.*

**RC1**: "Comment 13, P9 and following: the authors refer to their observations as nucleation; this is not correct since only >7 nm particles are observed. The use of "formation events" (eg P9 L6, L9) is also questionable, and an expression such as "the appearance of small particles" would in my view better describes the observations."

*Authors' response: The authors agree that formation events might be misleading with regard to TSMPS data in Fig. 6, as the measured aerosol particles are altered already. We have changed the expressions in the results and abstract, accordingly. P. 10, l. 3, p. 10, l. 8, p. 12, l. 15, p. 13, l. 12, p. 13, l. 17*
*In case of the expression "NPF" for the size of 5 and 7 nm, respectively, we refer to the authors' response to the short comment by Wolfgang Junkermann (here p. 13ff)*

**RC1**: "Comment 14, P9, L5-7: Based on the surface plot of the particle number size distribution (Fig. 6a), the increase of the particle concentration seems to be seen up to 30 nm (instead of 20 nm), and it is thus not correct to say that these events coincided with a decrease of the particle concentration in the range 20-50 nm."

*Authors' response: Thank you for the correction. We have changed the size range between 30 and 50 nm instead of 20-50 nm (p. 9, l. 30, p. 10, l. 4).*

**RC1**: "Comment 15, P9, L22-23: The sentence "The water vapour mixing ratio q increased during the day in the vertical distribution between 0 and 1500 m" is confusing and should be rephrased."

*Authors' response: The authors agree that the formulation causes confusion and the sentence was changed to:*
*The water vapour mixing ratio q reached maxima between the heights of 950 and 1210 m near the inversion layer.*
*The total maximum of $q=18.5$ g/kg was estimated (…) p. 11, l. 2-3*

**RC1**: "Comment 16, P9, Figure 8: "The vertical profile of turbulent kinetic energy… showed a strong connection with the structure of the ABL" (L24-25). What does this sentence mean? Also, regarding the analysis provided on P10, L6-8: it is quite complex to discuss/assess the effect of the parameters shown on Fig. 8 a and b since the vertical distribution of N7-12 is shown to vary faster than the time resolution of the model outputs.

*Authors' response: The authors agree that the description of the TKE is not sufficient. We intend to describe fast mixing processes in the well-mixed boundary layer that could be taken from the highest TKE rates. Yes, the model output is in 1 h interval, but in our opinion, it is still necessary to calculate the rates in order to explain the fast mixing processes seen in the CPC data. As it was described already, no other meteorological data was available in the verticality.*

**RC1**: "Comment 17, P10, L12: The expression "validate" might be too strong, as some of the profiles do not start from ground level, and thus do not allow for a direct comparison. This analysis is anyway very interesting, and, as mentioned previously, I would recommend to do it also for the first case study."

*Authors' response: Thank you for the comment. Indeed, the expression "validate" is too strong in this case. We have changed the corresponding lines (p. 11. l, 25).*
*For comparison, TSMPS data was estimated for the same size of $N_{7-12}$ (…)*
*In our mind, the integration of TSMPS data into vertical profiles would not benefit for the first case, as only larger particles were captured by the ground observation. The focus of Fig. 4 is on the impact of ABL processes and the progress in time.*

**RC1**: "Comment 18, P11, L3-20: It seems that some of the conclusions regarding the first case study reported in this section are different from those provided in section 3.1. In particular, the sentence on P11 L13-14 "however ground observations did not catch the newly formed boundary layer aerosol" does not suggest anymore the transport of particles downwards, and contrasts with the sentence from P8 L10-12 "the lifted layer of freshly formed particles was transported downwards, which can be further seen by the temporal appearance of the small particles of a few nm in diameter in the aerosol data at ground
level". Surprisingly, a possible connection with anthropogenic sources of particles is mentioned but NOx measurements shown in Fig 2b are not used to support this hypothesis. Also, the reference to Bianchi et al. (2016) should be reconsidered; the end of the sentence should be rephrased and the message clarified, as Bianchi et al. (2016) suggest that organic compounds from anthropogenic origin (and not CO) are involved in the NPF events they observe. CO is only used as a tracer for the anthropogenic origin/signature of the sampled air masses."

*Authors' response: Thank you for the detailed explanations in Comment 18 and Comment 19. The conclusion was revised in the discussion paper. Please, take into account the changes. Further, the correlation of NOx and sporadic events, as role for anthropogenic emission is considered in the revised form.*
*Thank you for the comment regarding the literature of Bianchi et al. (2016). Yes, indeed the outcome of the reference was stated wrong in our manuscript. The authors wanted to express that NPF occurs under more conditions as expected so far (p. 13, l. 1-4).*
*The study of Bianchi et al. (2016) at the Jungfraujoch site in the free troposphere might support the present observations that NPF occurs under more conditions as expected so far; NPF was not related to sulphuric acid formation, which is a common identifier for nucleation, instead NPF depended on high concentrations of organic compounds from anthropogenic origin.*

**RC1**: "Comment 19, P11, L21-33: While the authors consider the possibility for the particles formed aloft to have disappeared on L24, they assert on L25-27 that these particles are those observed close to the ground after vertical mixing. These sentences seem to be a bit contradictory, and the authors should be more careful in providing their conclusions, and use some formulations such as "might", "could", "suggest" … In particular, the analysis of the particles and CO2 fluxes used to support the conclusions seems to be uncomplete. Indeed, while CO2 fluxes show similar behaviour during the events observed at 10:10 and 16:10, particles fluxes show contrasting behaviour, being slightly positive during the first event and negative during the second one. This cannot be simply summarized as "During sporadic formation events, significant deposition occurred, taken from the negative values of particle fluxes. In

contrast, in between formation events, emission was observed, indicated by positive particle fluxes" (P9, L13-15). Also, as already suggested for the first case study, the authors should discuss more the disappearance of the particles."

*Authors' response: Thank you! We have completed the explanations of the particle fluxes (p. 10, l. 15-25).*

*In order to understand possible local processes in the current work, additional estimations of particle fluxes will be considered for the day. Fig. 7 displays particle fluxes taken from the EC (eddy-covariance) station that was deployed several metres away from the UAS operation and ground monitoring. Short lifetime and quick fluctuations of emission (shown in blue, positive particle flux) and deposition (shown in red, negative particle flux) are prevailing during the day. Significant deposition occurred with a maximal particle flux of $-26 \times 10^6$ $m^{-2}$ $s^{-1}$ at 12:30 UTC, taken from the 30 s averaging, that is in relation to the occurrence of ultrafine particles and the brake of ABL clouds. Deposition with a maximum of $-75 \times 10^6$ $m^{-2}$ $s^{-}$ was shown by Buzorius et al. (2001) during a clear nucleation event with particle growth rate, measured with cut-off sizes of 7 and 14 nm, respectively. Moreover, the study calculated the high frequency 87% of downwards fluxes during nucleation events that might lead to the assumption of local sources from upper layers so that not only sources near ground level should be considered. Possible causes for NPF in the ABL due to turbulence were as well previously suggested by Nilsson et al. (2001).*

**RC1**: "Comment 20, P11, L23: "The maximum … and increased rapidly while descending". This sentence is confusing, please try to rephrase."

*Authors' response: Thank you! Yes, the sentence is not correct and it was reformulated:*
*(…) decreased while descending (p. 13, l. 8)*

**Technical corrections**

**RC1**: "P3, L14: "high" should be replaced by "large"."
*Authors' response: Thank you for the comment. The authors have done the change, although the reference was not considered in the section (p. 12, l. 21).*

**RC1**: "P5, L10: "and" instead of "und"."
*Authors' response: Thank you for the hint of the typo! The authors have changed it accordingly, although the sentence was removed, anyway (p. 5, l. 30).*

**RC1**: "P6, and following sections: For clarity I would suggest to give all particle concentrations as N, and not dN/dlog(dp)."
*Authors' response: The authors agree that the expression might cause confusion. We have done the changes accordingly (p. 7, l. 9).*

**RC1**: "P6, L8: "of" instead of "on"."
*Authors' response: Thank you. Change was done (p. 7, l. 4).*

**RC1**: "P7, L11-12: I would suggest to change the expression "in the particle diameter". The authors can simply refer to N5, N10, N5-10, N390, N500 and N700 as those were defined before."
*Authors' response: The authors agree that it is easier to follow the results by the simple expression of N5, N10, N5-10, N390, N500 and N700, respectively, as the cut off sizes (CPCs) and size bins (OPC) were mentioned already in Section 2.1 p. 4, l. 1-12. The sentence was reformulated to "total aerosol particle number concentration $N_5$ (red line) and $N_{10}$ (blue line), given by the two CPCs, and the total aerosol particle number concentration with the three OPC channels of $N_{390}$ (pink line), $N_{500}$ (green line) and $N_{700}$ (turquoise line) …" (p. 8, l. 8-9).*
*However, we decided to maintain the description in Fig 4.*

**RC1**: "P7, L31: Please try to reformulate the expression "particles belonging to the diameter"."
*Authors' response: According to the previous comment, the authors used the simple term of $N_{390}$.*
*Therefore, the sentence was changed to: "At the same time, $N_{390}$ was evenly distributed below the inversion layer…" (p. 8, l. 28)*

**RC1**: "P8, L32: I would suggest to change the expression "in the particle diameter"."

*Authors' response: The authors have done the following change in order to avoid the expression "in the particle diameter": "in the size between 30 and 50 nm" (p. 9, l. 30).*

**RC1**: "P9, L1: "in the particle diameter range between 3 and 10 nm" instead of "of 3 and 10 nm"."
*Authors' response: Thank you! We have reformulated that sentence (p. 9, l. 32-33).*

**RC1**: "P10, L6: "790 m" instead of "790 nm"."
*Authors' response: Thank you for the comment. The typo was changed (p. 11, l. 19).*

**RC1**: "P10, L7: "$N_{7-12}$ decreased to the total aerosol number concentration…". This is confusing, please try to reformulate."
*Authors' response: The authors agree that the sentence causes confusion. Therefore, we have deleted the expression "to the total aerosol number concentration" (p. 11, l. 20).*

**RC1**: "P10, L16: "integration of a size scan"."
*Authors' response: The authors have done the change and the sentence is now complete (p. 11, l. 30).*

**RC1**: "P11, L7: "high concentrations of sulphuric acid"."
*Authors' response: The change was made on (p. 12, l. 21).*

**RC1**: "P17, Fig. 1: I would suggest to show the location of the station on the map, even if it said that the station is located in the domain centre."
*Authors' response: We will mark the domain centre in the new version.*

**RC1**: "P18, Fig. 2: the ticks on the x axis do not ease the reading of the times (10 intervals for 6 hours). Same applies to Fig. 6."
*Authors' response: Thank you for the hint! We will revise the ticks of the x axes accordingly.*

**RC1**: "P20, Fig 4: Logarithmic scales should be used for particle concentrations as it is very difficult to seen the variations of $N_{5-10}$ and $N_{700}$."
*Authors' response: Thank you for the suggestion. If the variations will be more significant to capture, we will change the scales in the new version.*

**Comments of referee #2 (RC2)**

"In summary this paper presents observations of non-surface layer new particle formation under cloudy conditions and near surface vertical transport of newly formed particles. Observations in the vertical were made using the UAS system ALADINA in the region of the TROPOS research site near Melpitz: TROPOS facilities providing surface observations. 53 hours of flight data was available spanning a period 23 days between October 2013 and July 2015. Of the available data two periods were selected as case studies.

Although the paper is well structured and presented the quality of the English is poor making for a difficult read and confusion as to what point the author is trying to convey – there are far too many corrections to list here. There is also a tendency for the author to uses emotive\subjective phrases for example: "All in all, the research site Melpitz of TROPOS offers a great potential for observing NPF within the ABL and inter-comparison of airborne data with monitoring at ground." These are major failings and ones that need to be addressed: if the author is not a native English speaker it is recommended that they find somebody who is to assist in editing.

The work presented is unique and will be of great interest to the community but there needs to be a substantial "tightening up". The paper gives the impression of slackness and carelessness in experimental techniques, discussion of relevant processes, and how conclusions have be arrived at. At this stage and in this form I would not recommend publication but I would encourage the authors to re-submit after revision."

**Author's response to RC2**

The authors acknowledge the anonymous referee #2 for the support in review process. The given comments will help to "tightening up" the manuscript in its present form. As it was assumed already, the authors are not native speakers, but will take into account the editorial support, if still necessary after revision. In most cases, the authors will refer to previous comments to referee #1 and to the detailed answers to Wolfgang Junkermann. So we therefore ask you for understanding that most of the responses are rather short.

Please, find attached the point-to point response and the marked-up manuscript version including the changes (removed is marked in red and blue is coloured in blue).

**Point-to-point response:**

**RC2:** "1. The authors site a number of papers concerning the experimental set up at both the field site on for the UAS. It would be expected for a basic discussion of both these setups to be included in the paper and a discussion of the various factors that will effect measurement uncertainty and what has been done to mitigate them. This is very important when it comes the particulate measurement as inlet loses and transmission losses can seriously bias a measurement: for example is the fact that you are not seeing particle above a certain size because they are not there in the first place or because they have all been lost before reaching the sensors."

*Authors' response:* The authors agree that this is a major point. However, we decided not to repeat the technical part of the measurements, as it was clarified in the previous paper by Altstädter et al., 2015, already. In addition, the results in Platis et al., 2016 showed a good agreement between UAS and surface data (seen in NAIS). The focus of the current publication was only on the results. Nevertheless, we have extended the section (see comments on p. 1).

**RC2:** "Basically have you characterised the both the ground and UAS sampling and measuring systems. If you have then discuss it and the implications. If this has not been done you would be advised to do so."

*Authors' response:* Yes, we did several comparisons with the UAS and ground measurement, sampled at the same aerosol inlet at ground. One event was shown (for instance), in Bärfuss et al., 2018. The system was redesigned in the meteorological set up, but the aerosol instrumentation is still valid. The uncertainty of +/-20 % of the CPCs was verified by comparison of N7-12 with TSMPS data in the same diameter size. Please, take into account our remarks at the very beginning of the authors' response on p. 1. (…) The following lines were added to the new manuscript (p4., 16-18):

The laboratory results were confirmed by Bärfuss et al. (2018) during field studies in Melpitz. The UAS was placed at the same aerosol inlet as ground monitoring and N7-12 coincide with ground data in the same particle size within the deviations of +/+20% during sampling period of 1.5 h.

**RC2:** "2. There is also the point regarding humidity and this applies to both the ground and UAS measurements. You are measuring the humidity of the ambient air sample upstream of your sensors. If that air mass is warmed (resulting in its RH reducing) or purposefully dried then this will impact on particle size. There is a need to be exact as to what the conditions are at the point of measurement and to indicate what effect this might have."

*Authors' response:* This is correct, the relative humidity needs to be considered for size-resolved measurements without drying the sample flow. Ground based aerosol measurements at Melpitz are always performed under dry conditions according to Wiedensohler et al. (2012), that means the measurements are performed below 40% rH in the

*sample flow. Thus, further heating does not influence the particle size anymore. But for measurements on ALADINA rH may have an influence. In general, a change in rH does not alter the particle number concentration but it may change the particle diameter. This could influence the lower cut of the CPCs in comparison with dry measurements and the upper and lower diameters of OPC size channels. In general the hygroscopic growth factors are smaller for smaller diameters. For a typical aerosol composition in Melpitz a 10 nm-particle has a growth factor of approximately 1.3 at 90% rH and a 500 nm-particle a growth factor of 1.8. This is the maximum difference in diameters for our measurements if we compare dry and ambient measurements. Since the aerosol flow is also heated while entering the inlet system of ALADINA, the real deviation will be smaller.*
*As mentioned above, the hygroscopic growth does not change the number concentration measured using ALADINA.*

**RC2:** "3. The need to be exact also extends to terms you are using. You refer to size but what do you actually mean: radius, diameter, and are these optical, mobility or aerodynamic equivalent values. This needed to be exact also extends to the gas phase measurements: are the concentration in ambient air or dry air? What are the uncertainties in these measurements?"

*Authors' response:  The gase phase measurements are in ambient air.  In case of the OPC we talk about optical diameters. For the CPC cut-off detection a DMA is used, thus mobility diameter.*
*The uncertainties were given in Section 2.1 on p. 4:  CPCs (TSI) +/- 20% and OPC (Met One Instruments) +/- 15%*

**RC2:** "4. With regard to your CPCs. In the first instance you are using two units: one with a lower cut of 5 nm (diameter?) and one with a lower cut off of 10nm (diameter?) while in the second you are using two units: one with a lower cut off of 7nm (diameter?) and one with a lower cut off of 12nm (diameter?) and on page 4 lines 1 – 6 this is clear. Lines 10 – 14 on page 7 this is not the impression the text gives. This is what you say: "total aerosol particle number concentration measured with two CPCs in the particle diameter of 5 nm (red line) and 10 nm (blue line),". What I think you are talking about is the total aerosol concentration measured by the two cpc: red line cpc with 5nm lower cut off, blue line the cpc with a 10nm lower cut off.
On line 12 of the same page you say: "OPC in the particle diameter of 390 nm (pink line), 500 nm (green line) and 700 nm (turquoise line)". An OPC has specific measurement bins that will have a centre value of particle diameter and a specified width – you need to add the width of the bin
to this kind of statement and state explicitly that you are using the bin centre value.

*Authors' response: Thank you for the comment, however, the size bins were described in the previous section. Please see the description of the OPC on p. 4, l. 19-24. For simplification, the bins were named to N390 for size range between 390 and 500 nm, N500 for 500 to 700 nm and N700 stand for 500 to 700 nm, respectively.*

**RC2:** "5. Your figures show no error bars nor is there any reference in the captions: this needs to be addressed. On the subject of figures, although I am not colour blind many of your readers may be. It is good practice to choose colour schemes that those who are colour blind can interpret."

*Authors' response: Thank you for the hint! We will change the colour scale in a different way. Please take into account the given references of the instrumentation (marked up in the discussion paper).*

**RC2:** "6. In your discussion of the use of the LES you use the term "closure": however what you are actually doing is using the model output to substitute for missing observations.

*Authors' response: Yes, indeed. Please, see the comments to the first referee. We have changed the misleading expression accordingly (p. 1, l. 20).*

**RC2:** "Again the need for exactness but there is also the question of the validity of this approach.
What evidence can you present that would indicate that the LES has accurately representing the conditions at the site. Presenting a comparison the model data with flights where you have full observations would be advised."

Authors' response:  *Again, we refer to the first comments of the authors' response, as the data set was not the same during the both measurement days. The comparison of LES model with UAS data is not the scope of the paper, but we will consider another measurement day for the overview paper of the MelcCol 2015 campaign. Further, the output considers data from the German weather station in Torgau (3 km away from the site) and radiation measurements from the Melpitz site. Additional information was provided in the text in Section 2.3 (p. 6, l. 19-23).*
*"The utilised model set-up is similar to the one described in Heinze et al. (2017), and it includes large-scale forcing tendencies due to advection of heat and water vapour, subsidence from COSMO* (Consortium for Smallscale

*Modeling, Baldauf et al., 2011) reanalysis (Heinze et al., 2017); direct, diffuse and terrestrial radiation (1 min averages) directly measured at the field site and soil data (temperature and moisture) from the German weather service (Deutscher Wetterdienst, DWD) station "Klitzschen bei Torgau", which is located 3 km away from the Melpitz field site."*

**RC2:** "7. Lidar and ceilometer. You put a deal of emphasis on the importance of these observations without the necessary explanation and discussion of the observation themselves. What are you actually looking at what is the implication to the interpretation arising from the change of laser wavelength?"

*Authors' response: The authors intended to obtain a full daily analysis of the cloud coverage and aerosol layers, as measurement flights were only operated below clouds and during sunlight. Besides, the data and lidar was used to affirm the existence of clouds during the UAS observations. For detailed description of the backscatter signal, we refer to p. 5, l. 25-29.*

**RC2:** "8. In your paper introduction you introduce many concepts and make many statements backed up by a paper reference – you could really do with expanding this by adding brief textural explanations."

*Authors' response: Thank you very much for the hint. The authors agree that the introduction is short in its present form. However, we wanted to summarise the so-called favoured conditions for NPF that are stated in the different literature from "(e.g. Wiedensohler et al., 1996; Keil and Wendisch, 2001; Birmili et al., 2003; Kulmala et al., 2004; Hamed et al., 2010; Hamburger et al., 2011)".*

**RC2:** "9. You introduce "gravity waves" but do not provide an explanation as to why this is relevant and the process that are at play."

*Authors' response: Thank you for the comment. Please, note the answer to referee #1 (p. 3, Comment 3 of authors' response).*

**RC2:** "10. Use of the term inter-comparison. This is a tautology and a very common mistake
– simply use the term comparison."
*Authors' response: Thank you very much for that hint! We have changed it accordingly (p. 5, l. 9).*

**Short comments by Wolfgang Junkermann (SC1)**

"The paper shows some interesting data on elevated ultrafine particle layers under total overcast conditions obtained, using an unmanned aerial vehicle, the UAV ALADINA, flying vertical profiles through the planetary boundary layer and lower free troposphere.

The occurrence of ultrafine particles under such meteorological conditions would not be expected following the general description of new particle formation observed elsewhere.

The authors nevertheless, claim a clear identification of a new particle formation event. Typically for the production of ultrafine particles via new particle formation or, with the old expression, gas-to-particle conversion, shortwave radiation is suggested to be required as an initial step to convert first sulphur dioxide into sulphuric acid, which than reacts in the atmosphere often with ammonia or similar compounds producing initial aerosol clusters. This is well established in a large number of publications, see for example for a review Kulmala et al, 2011, and Kulmala et al, 2013 for an analysis of the expected particle generation process. Growth occurs then via (EL)VOC, further sulphuric acid water etc. . The data shown here were measured under conditions without sufficiently high UV radiation levels to produce sulphuric acid in the atmosphere, thus requiring a different production process. Such an occurrence of ultrafine particles under totally overcast conditions has been observed several times and was reported for example over Germany (Junkermann et al, 2016), a paper that is cited in the manuscript. Here these particles were apportioned to primary industrial sources

burning fossil fuel. Hence, as several of these elevated sources are located around Melpitz the observations are not really surprising. Anyway, the data analysis leaves several questions"

*Authors' response:* *First of all, the authors acknowledge Wolfgang Junkermann for his detailed comment and discussion. However, we disagree in some points. Please, find attached the corresponding answers.*

**SC1**: "1) The manuscript states that NPF occurrence with subsequent downward transport

was clearly identified. The question: What does 'new' mean within this context? Was the particle formation (cluster production and subsequent growth) really been observed?"

How can a nucleation (gas to particle conversion) be identified with instrumentation that is not able to measure the initially formed particles? From the two size fractions the size of the major nucleation or Aitken mode cannot be estimated. It could be anything larger than 12 nm and only shows, that the distribution contains some particles in the range below 10 or 12 nm.

A gas to particle (NPF) event however, should have as a minimum requirement initially the majority of particles in the lowest size bins of the distribution. The number of particles in the size between 7 and 12 nm (or N7-12) has then to be higher than the number of particles > 12 nm before and at the time the growing nucleation mode is passing this size range. Here, within the data presented the difference between the size bins is marginal, compared to the overall number concentrations indicating rather an aged air mass. Occasionally the N10 is even higher (measurement uncertainty?) than N5., in Fig. 4b and 4c N10 is missing at all."

*Authors' response:* *This paragraph might be misleading. The UAS delivers a snapshot of the current state and not the formation process itself, due to missing data of chemical gas phase and the use of CPCs models with a lower cut-off size of at least 5 nm. The authors did not state at any time, that sources for NPF were investigated by the set up. Besides, this is not the goal of the publication, as it is only to show the evidence that NPF occurs more often as expected so far and would not be taken into account by pure surface data. We use the expression "new particle formation" or NPF for particles in the size range of 5 to 10 nm (or 7 to 12 nm), in contrast to larger, aged aerosol particles.*

*Now, the answer to this comment is more detailed, because all the following comments are linked to this one: Yes, that is correct, NPF consists of two steps, but the result is the occurrence of small particle. That means, if we observe the result, we can conclude that particles were formed by gas-to-particle conversion, because this is the only process to produce such small particles in the atmosphere. We observed an increased number concentration in the size range below 12 nm. The only way to produce such small particles is gas-to-particle conversion, thus we know that NPF took place without measuring the clusters itself. This is a well-established method to investigate and identify NPF events.*

*We think the events observed here are locally restricted and not homogeneous over larger regions. This is because in particular the clouds create inhomogeneities in the intensity of solar radiation reaching the ground. Thus, we cannot expect a homogeneous development of the boundary layer. On the other hand, clouds and their surrounding regions as well as inversion layer are regions with large gradients in various parameters. These gradients in combination with local turbulence may create strong nonlinearities being a favourable environment for new particle formation which has been discussed theoretically by Easter and Peters (1994) and shown in- situ by Wehner et al. (2015). The dynamics of such an event being locally restricted to small regions are completely different and cannot be explained by characteristics of a large-scale event and relation to air mass.*

*The small NPF events may occur suddenly, followed by fast growth but also fast dilution with surrounding air. If the stadium of dilution starts, the number concentrations decrease immediately and if an event is detected in this stadium the number concentration below 12 nm is less significant.*

**SC1:** "2) Following Kulmala et al, 2013, it takes several, ~about five or more, hours for a new particle to grow into the lowest size range of the particles that can be observed with the instruments on ALADINA. Observation of particles at 08:00 UTC thus requires a production at ~03:00 UTC or even earlier during the night. These particles have to be generated in a different area, a minimum of five hours upwind, which is according to HYSPLIT at least 70 km. Growth rates are normally smaller or even zero at night. Thus the source could be even further away."

*Authors' response: Growth rates can be much higher in combination with turbulence. Small fluctuation in combination with large gradients may create local supersaturations of potential precursor gases for nucleation and growth. The lifetime of such events could be extremely short as shown in Wehner et al. (2015), where Markku Kulmala has been co-author and did not see any conflict to his earlier studies. The formation and growth of these small and short events is different to those happening over larger areas and longer time scales. The latter ones are often combined with the typical banana-shaped measurement on ground and occur over larger areas.*

**SC1:** "3) Where is the production area of the particles, which are 'clearly' identified as originating from NPF, according to HYSPLIT ~70-100 km for 5 h?"

*Authors' response: At this point, we refer to the comment above. The NPF process here is different and occurs on a much shorter time scale.*

**SC1:** "4) Why are the meteorological and atmospheric chemical conditions favorable for new particle formation > 70-100 km upwind, during the night and in the elevated layer?
Where do the precursor chemical compounds come from? What could be the initial step for nucleation cluster generation at night?"

*Authors' response: Again we would like to refer to the previous points: NPF did not start hours ago, because the growth rates are much higher in this case. These events are very local, thus we assume the particles have been formed right in the measurement region and less than one hour ago.*

**SC1:** "5) Is there probably a source for ultrafine particles upwind that could explain the results?
HYSPLIT shows that the air mass has been close to the German-Polish border on the evening of April 3. Here we have at least three power stations that are sources for primary particles in the respective size ranges as well as additional large amounts of sulphur dioxide, ammonia (from the SCR cleaning process) and also internally produced sulphuric acid (see also Junkermann et al, 2016). Size distributions, independence on time of the day (Junkermann and Hacker, 2015) and laboratory results confirm primary emission of 'New' particles (Brachert et al, 2013). Such particle production does not require UV radiation or OH radicals. A quick HYSPLIT analysis: For June 21 the winds in the altitude of 600 m above ground come from the northwest passing the power station Buschhaus as the next possible candidate. Further upwind (up to 24 h) along the backtrajectory the industries of Groningen (NL) are located. For westerly to southwesterly winds in the PBL the power stations Schkopau and Lippendorf as well as the refinery Leuna would be possible sources for primary ultrafine particles and further precursors over Melpitz (see also Platis et at, 2015). For size distributions and source strength of such sources see Junkermann et al, 2011; 2016."

*Authors' response: If there would be a source of ultrafine particles upwind of the site, the particles would grow fast during the transport. The authors doubt that ultrafine particles are directly emitted by a combustion source. Modern burning facilities using oil or natural gas emit gases which may form ultrafine particles directly in the exhaust due to cooling and condensation. This happens also in the exhaust of cars and these ultrafine particles can be measured already close to the tailpipe. However, they were not emitted as ultrafine particles, they were formed afterwards. Combustion processes produce larger particles with diameters above 50 nm and no ultrafine particles.*
*Furthermore, ultrafine particles in Melpitz and elsewhere have been characterized to be volatile. This means they consist of condensable material and were not formed during e.g. a combustion process.*

*Furthermore, if the ultrafine particles observed in Melpitz would have been formed far away, they would not appear as single bursts. Such small structures with sharp gradients to the environment would dilute very quickly within the order of minutes in maximum. Thus, it is simply not possible that these particles ware produced elsewhere and transported to the site.*

**SC1:** "6) Page 9, line 20, atmospheric boundary layer conditions were unfortunately not available from the UAS. Instead a model is used to characterize the ABL. However, this model is not able to reproduce the measured vertical structure (Fig. 9). Why are no model data shown for 07:00 to 08:00, the time window before the

aircraft measured the pronounced vertical profile? Lateron the PBL rapidly mixed. The vertical structure was visible only within the first 15 minutes of the model 1 h time window."

*Authors' response: The authors agree that the presence of different data set of the two measurement sites was not clear in the old version of the manuscript. Please, take into account the comments on the first page of the authors' response. But, we do not fully understand the questions (in point 6). Fig. 8 shows modelling profiles of potential temperature, mixing ratio and TKE for each hour from 6 to 16 UTC. In Fig. 9, aerosol profiles obtained with ALADINA and ground-based instrumentation is shown. However, aerosol profiles are not derived from the model.*

**SC1:** "7) Fine particles: particle number concentrations > 390 nm on April 4 are fairly high.
We can compare to particle numbers measured with a GRIMM 1.108 (fraction > 400 nm, second size bin) over Kathmandu (missed approach to Kathmandu international airport) in January 2014, to data gathered during flights in the extremely dusty Mexico City basin during Milagro 2006 or to data from the heavily polluted Po-Valley (QUEST, 2004). The number concentrations there were only about half or less of what has been seen over Melpitz. Under these conditions the condensational sink (CS) would be extremely high and according to most of the published literature a NPF event would be very unlikely. How is this in agreement with the summary / abstract statement that New Particle Formation has been clearly observed? The summary claims: Further, the NPF event was 5 linked to a condensation sink of larger particles belonging to the accumulation mode at the same altitude. That's not in agreement with the fig. 4, the layers are clearly separated."

*Authors' response: The OPC number concentration in the size class 390 – 500 nm ($N_{390}$) was between 200 and 250 $cm^{-3}$, which is not extremely high. To validate the vertical measurements we can take the number size distributions measured on ground and integrate over the corresponding size range. Additionally we have to consider that on ALADINA the sample flow was not dried, while on ground, the number size distribution was measured below 40% rH behind a dryer. Thus, if we want to compare both measurements, we have to consider the hygroscopic growth of the particles. On April 4 between 9 and 14 UTC rH varied between 70 and 90% corresponding to a growth factor between 1.17 and 1.25. Using this, the diameters of $N_{390}$ move to smaller sizes and the size range for dry particles would be $320\pm10$ nm to $410\pm10$ nm. Integration over this size range results in values between 230 and 320 $cm^{-3}$ during the observed period. This fits well within the uncertainties to the results from vertical measurements. The layers are not separated in Figs. 4a and 4b. We mentioned the occurrence of a second layer in Fig 4c (p. 8, l. 32-33) that did not coincide with the first layer above/near the inversion layer.*

**SC1:** "8) Fig. 9 and page 11, line 22/23: 'The maximum of total aerosol particle number concentration was 1.6_105 cm-3 at the altitude of 420ma.g.l. and increased rapidly while descending'. Not in agreement with the figure. The number concentration below 420 m actually is always lower than the maximum at 420 m and decreased between 08:13 and 08:26 UTC. Probably also a result of changing wind directions (~15 deg within 2 hours)"

*Authors' response: Thank you for the comment. The authors agree that the sentence is wrong and have changed it accordingly (p. 13, l. 8).*

**SC1:** "Summarizing: The paper does not really show evidence or proof for gas to particle conversion respective new particle formation. Though interesting and good to identify ultrafine particle layers the instrumentation used is most probably not appropriate to detect such particle formation events. An SMPS or NAIS would be necessary. The results are contrary to the general literature about new particle formation and unfortunately, there is no attempt to analyze where the particles observed are produced or originating from. However, the measurements would be in agreement with emissions from a few welll known 'continuous generators for ultrafine particles' e.g. power stations /refineries and the final statement that a thorough meteorological analysis (but, not only in the vertical) is necessary to understand ultrafine particle behavior in the PBL is strongly supported."

*Authors' response: The authors did not intend to state new particle formation production, as it is well known that the CPCs are not valid for the smallest diameters of a few nm. However, as it is stated in the title, the authors wanted to show, that new particle formation was observed under conditions that are in general not favourable for NPF. Besides, you can take from the title that only observations are stressed and not the initiation of NPF.*

**Authors' changes**

Please not additional changes that were done by the authors. For instance, some literature was added/revised during the review process.

- p.1, affiliations: *Now at: Laboratory for Air Pollution/Environmental Technology, Empa, Swiss Federal Laboratories for Materials Science and Technology, Dübendorf, Switzerland
- p.1, l. 3-4  Inserted affirmed by ceilometer and lidar data
- p.1, l. 9 Inserted "mixed and mainly"
- p.1, l. 12 changed two cases into "two measurement days"
- p.5, l. 19" was expected" instead of were expected
- p.15, l. 10-12 Inserted new literature for COSMO (Baldauf et al., 2011)
- p.15, l. 13-15 During review process, the publication of Bärfuss et al. (2018) was accepted.
- p.15, l. 30-31 New literature of Buzorius et al. (2001)
- p.16, l. 1-3 Added new literature of Engelmann et al. (2016) for polly
- p.16, l. 7-9 During review process, the publication of Größ et al. (2018) was revised.
- p.16, l. 19-20 Inserted new literature of Heese et al. (2010).
- p.17, l. 20-21 New reference of Pal et al. (1992)
- p.18, l. 20-21 New literature of Wiegner and Geiß (2012)
- p.19, l. 4 changed typo of resolution

[revised manuscript text omitted]

The aircraft Carolo P360 "ALADINA" (Application of Light-weight Aircraft for Detecting IN-situ Aerosol) was designed and developed for atmospheric research in order to investigate the vertical and the horizontal aerosol distribution depending on atmospheric boundary layer properties.  The aircraft design and instrumentation's set up were introduced in Altstädter et al. (2015) and supported by the results of Platis et al. (2016). During review process, the aircraft was re-engineered and we refer to the newest study of Bärfuss et al. (2018) who presented a profound overview of ALADINA's flight operation, data procedure and the current payload that  is equipped with aerosol instrumentation and meteorological sensors with high temporal resolution.

The total aerosol particle number concentration is derived by two Condensation Particle Counters, CPCs (model 3007, TSI Inc., St Paul, USA), with different lower threshold diameters. In the first case study here, the cut-off sizes were 5 and 10 nm, respectively. The difference in the particle number concentrations of both CPCs ($N_5$ and $N_{10}$), in the following referred to as $N_{5-10}$, is used for the number concentration of freshly formed particles. During the second case study, the lower threshold diameters of both CPCs were 7 and 12 nm ($N_7$, $N_{12}$; $N_{7-12}$), respectively. The CPCs were characterised to measure within an uncertainty of $\pm 20\,\%$ with a fast response time of 1.3 s. The laboratory results were confirmed by Bärfuss et al. (2018) during field studies in Melpitz. The UAS was placed at the same aerosol inlet as ground monitoring and $N_{7-12}$ coincide with ground data in the same particle size within the deviations of $\pm 20\,\%$ during the sampling period of 1.5 h.

In addition, an Optical Particle Counter, OPC (model GT-526, Met One Instruments Inc., Washington, USA), is installed and measures the size distribution of aerosol particles with six channels from 0.39 to 10 µm (ambient) in particle diameter with an uncertainty of $\pm 15\,\%$ and a temporal resolution of 1 s. Here, the aerosol particle size distributions are analysed in the size windows between 390 to 700 nm, as larger particles were not relevant in the study due to minimal appearance. In the following, the particle size distributions of the three channels refer to the total aerosol particle number distribution in the size range between 390 and 500 nm ($N_{390}$), between 500 and 700 nm ($N_{500}$) and 500 to 700 nm ($N_{700}$).

The meteorological instruments are mounted at the tip of the aircraft nose next to the aerosol inlet. The sensor package consists of one five hole probe for measuring the three-dimensional wind vector with a temporal resolution of up to 40 Hz and wind speed with an accuracy of $\pm 0.5\,\mathrm{m\,s^{-1}}$ (Wildmann et al., 2014a). The fast temperature sensors have a resolution of 10–20 Hz with an accuracy of $\pm 0.1$ K (Wildmann et al., 2013). Additionally, one humidity sensor is integrated that probes the water vapour content with a response time of 1.5 s with $\pm 3\,\%$ RH accuracy (Wildmann et al., 2014b).

**2.2 Research site Melpitz and available instrumentation during experiments**

[revised manuscript text omitted]